# STRUCTURED FLOW AUTOENCODERS: LEARNING STRUCTURED PROBABILISTIC REPRESENTATIONS WITH FLOW MATCHING

**Yidan Xu, Yixin Wang & Long Nguyen**
Department of Statistics
University of Michigan
`{yidanxu,yixinw,xuanlong}@umich.edu`

## ABSTRACT

Flow matching is a powerful approach for high-fidelity density estimation, but it often fails to capture the latent structure of complex data. Probabilistic models like variational autoencoders (VAEs), on the other hand, learn structured representations but underperform in sample quality. We propose Structured Flow Autoencoders (SFA), a family of probabilistic models that augments graphical models with conditional continuous normalizing flow (CNF) likelihoods, enabling flow-matching-based structured representation learning. At the core of SFA is a novel flow matching objective that explicitly accounts for latent variables, allowing joint learning of the CNF likelihood and posterior. SFA applies broadly to graphical models with continuous and mixture latents, as well as latent dynamical systems. Empirical studies across image, video, and RNA-seq data show that SFA consistently outperforms VAEs and their structured extensions in generation quality, representation utility, and scalability to large datasets. Compared to generative models like latent flow matching (LatentFM), SFA also produces more diverse samples, suggesting better coverage of the data distribution.

## 1 INTRODUCTION

Generative modeling has become a foundational pillar of modern machine learning, offering powerful tools for capturing complex data distributions and generating high-quality samples. Among recent advances, diffusion models (Ho et al., 2020; Nichol & Dhariwal, 2021; Song et al., 2020; 2021b;a; Austin et al., 2021) and flow-based methods (Lipman et al., 2022; Liu et al., 2022; Gat et al., 2024; Tong et al., 2024; Isobe et al., 2024) have shown remarkable performance as neural density estimators, excelling at likelihood estimation and high-fidelity sample generation. In particular, flow matching has emerged as a scalable and efficient approach, aligning vector fields of probability paths using optimal transport principles, enabling efficient and scalable generative modeling with exact likelihood evaluation (Lipman et al., 2022; Liu et al., 2022; Gat et al., 2024).

Despite their success in generation quality, neural density estimators like flow matching often fall short in *structured representation learning*, failing to capture or expose the rich latent structures underlying complex data. This limitation is especially salient in scientific and structured domains such as computational biology, where interpretable low-dimensional representations are essential for downstream tasks, analysis, and control. Recent work has revealed both empirical evidence of implicit low-dimensional structures in pretrained diffusion models (Wang & Vastola, 2023; Chen et al., 2024) and theoretical guarantees of their adaptivity to such structures (Wang et al., 2024; Li & Yan, 2024). However, these models neither explicitly model latent structure during training nor produce readily interpretable representations, limiting their utility beyond sample generation.

In contrast, probabilistic latent-variable models such as variational autoencoders (VAEs) (Kingma & Welling, 2013; Johnson et al., 2016) are explicitly designed to capture latent structure through probabilistic encoder-decoder architectures. These models learn structured probabilistic representations that can be leveraged for conditional generation and downstream tasks. However, VAEs typically underperform in data modeling and generation fidelity compared to modern flow-based

Table 1: Comparing generated samples to data samples with $W_1$ metric (Earth Mover's Distance). $W_1$ metric is evaluated with samples from marginal data distribution $p(\boldsymbol{x}_1)$ and that generated from $\tilde{p}_1(\boldsymbol{x}_1) = \int p_1(\boldsymbol{x}_1|\boldsymbol{z}_1)q_1(\boldsymbol{z}_1)d\boldsymbol{z}_1$. SFA and FM achieve comparable performance on marginal density estimations.

|  | VAE | VampVAE | Mixture-SVAE | FM | LatentFM(w/VAE) | SFA | Mixture-SFA |
|---|---|---|---|---|---|---|---|
| $\hat{W}_1 \downarrow$ | 0.119 | 0.081 | 0.457 | **0.025** | 0.496(0.145) | **0.024** | 0.046 |

(a) Truth    (b) FM    (c) LatentFM    (d) LatentFM VAE

(e) VAE    (f) Mixture SVAE    (g) SFA    (h) Mixture SFA

Figure 1: Generated samples on the Pinwheel dataset with 5 clusters. Color in (a) indicates class membership, which is not provided during training. Color in (c) indicates the latent distribution learned via deterministic autoencoder. Color in (d)-(h) indicates the generated posterior value $\boldsymbol{z}_1 \sim q(\boldsymbol{z}_1|\boldsymbol{x}_1)$ given the generated sample $\boldsymbol{x}_1$. We use a continuous latent variable $\boldsymbol{z}$ in (c),(d),(e),(g); and a mixture latent variable in (f),(h).

models, limiting their utility in high-resolution or diverse generative tasks. This gap in generative fidelity also raises concerns about the reliability and expressiveness of their learned representations. This performance gap raises the question: *Can we build models that retain the structured latent representations of VAEs while achieving the high fidelity and scalability of flow matching?*

**Main idea.** We propose *structured flow autoencoders (SFA)*, a new family of probabilistic models that augments graphical models with conditional Continuous Normalizing Flow (CNF) likelihoods. This family aims to combine the strengths of both approaches: the high-fidelity data modeling of neural density estimators and the structured representation learning capabilities of graphical models.

We motivate with a simple latent variable model where continuous latents $\mathbf{z} \in \mathbb{R}^p$ generate observations $\mathbf{x} \in \mathbb{R}^d$, with $0 < p < d$:

$$\mathbf{z}_i \overset{i.i.d.}{\sim} p(\mathbf{z}), \qquad \mathbf{x}_i|\mathbf{z}_i \overset{ind.}{\sim} p(\mathbf{x}|\mathbf{z}). \tag{1}$$

This standard latent variable framework enables structured representation learning through the posterior $p(\mathbf{z}|\mathbf{x})$. To enable high-fidelity data modeling in SFA, we parametrize $p(\boldsymbol{x}|\boldsymbol{z})$ using conditional CNFs, achieving the expressivity of modern neural density estimators while maintaining structured latents. However, both the likelihood and the posterior are no longer available in explicit forms. To address this challenge, we propose the *Structured Conditional Flow Matching (SCFM)* objective, a training objective that jointly learns both the conditional flow $p(\mathbf{x}|\mathbf{z})$ and an approximate posterior $q(\mathbf{z}|\mathbf{x})$. Unlike standard flow matching that only models $p(\mathbf{x})$, SCFM explicitly accounts for the conditional structure $p(\mathbf{x}|\mathbf{z})$ and posterior $p(\mathbf{z}|\mathbf{x})$. As illustrated in Figure 1, this decomposition enables SFA to capture interpretable latent variables while maintaining high-fidelity generation, providing structured representation learning unavailable in standard flow matching.

**Contributions.** (1) We introduce Structured Flow Autoencoders (SFA), a family of generative models that augments graphical models with conditional Continuous Normalizing Flow (CNF) likelihoods. SFA bridges the gap between high-fidelity neural density estimation and structured representation learning, improving upon both VAEs and latent flow-based models. (2) We propose Structured Conditional Flow Matching (SCFM), a novel training objective that extends flow matching to explicitly incorporate latent variables. SCFM explicitly learns the conditional probability flows in

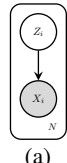 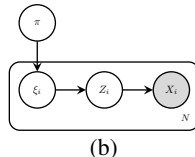 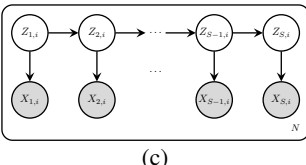

(a)  (b)  (c)

Figure 2: Examples of graphical models incorporated into SFAs: (a) latent continuous variable model; (b) latent finite mixture model; (c) latent linear dynamical system.

the graphical model while preserving the marginal density information. SCFM enables joint learning of the likelihood and posterior, supporting both generative modeling and structured representation learning within a unified framework. (3) We demonstrate the flexibility of SFA across diverse domains, including image, video, and RNA-seq data, and modeling scenarios with continuous, finite mixture, and dynamical latent variables. SFA achieves high-fidelity sample generation, increased sample diversity, and enhances structured representation learning, while remaining computationally efficient on high-dimensional datasets.

**Related work.** Simultaneous high-fidelity generation and structured representation learning has been an important task (Grathwohl et al., 2018; Mittal et al., 2023; Dao et al., 2023; Davtyan et al., 2023), drawing particular interest in scientific domains (Bashiri et al., 2021; Xu et al., 2023; Kapoor et al., 2024). Variational autoencoders (VAE) (Kingma & Welling, 2013) is one such probabilistic model that learns both generative model $p(\boldsymbol{x}|\boldsymbol{z})$ and inference model $p(\boldsymbol{z}|\boldsymbol{x})$ simultaneously, typically using neural networks to parameterize exponential families. Grathwohl et al. (2018); Chen et al. (2020) extended the VAE to families of normalizing flows, which improves the flexibility of density estimation together with latent space learning. While appealing, VAEs fall short of modern generative models in data modeling fidelity.

Recent work has explored combining neural density estimators with encoder-decoder frameworks (Mittal et al., 2023; Dao et al., 2023; Davtyan et al., 2023; Vahdat et al., 2021), typically mapping observations to low-dimensional latent spaces where the latent marginal distribution is learnt via flows or diffusion models before decoding back to observation space. While these neural prior methods excel at dimensionality reduction, they often constrain encoders and decoders to simple parametric families (e.g., Gaussians) that inadequately capture complex data distributions. Our approach differs by making the entire likelihood and posterior flexible through conditional flows while incorporating structured latent dependencies. Another line of work, including Wang et al. (2023); Preechakul et al. (2022), focuses on modeling the likelihood with flexible diffusion models while maintaining simple Gaussian posteriors. However, these approaches lack mechanisms for structured dependencies between latent variables, limiting interpretability in complex domains. Most closely related to our work, structured variational autoencoders (SVAEs) (Johnson et al., 2016; Lin et al., 2018) incorporate graphical model structure into VAEs to capture hierarchical dependencies. However, SVAEs are constrained by parametric assumptions that limit expressiveness. In addition, extending SVAEs to more expressive density models like CNFs faces significant challenges: direct extensions suffer from numerical instability and computational inefficiency due to the need of likelihood evaluation at every training step (Liu et al., 2022). Our flow matching approach circumvents these issues while enabling both structured representations and expressive data density modelling.

## 2 PRELIMINARIES: FLOW-BASED GENERATIVE MODELING

We start by reviewing continuous normalizing flows and the flow matching learning objective, laying the groundwork before introducing structured flow autoencoders (SFAs).

**Notations.** We follow the notations in Lipman et al. (2022) and denote the time indexed vector field by $v(\cdot, \cdot) : [0, 1] \times \mathbb{R}^d \to \mathbb{R}^d$ and equivalently, $v_t(\cdot) : \mathbb{R}^d \to \mathbb{R}^d$ for $t \in [0, 1]$. The path of probability densities is denoted by $p_t(\cdot) : \mathbb{R}^d \to \mathbb{R}^+$, and the flow $\phi_t(\cdot) : \mathbb{R}^d \to \mathbb{R}^d$ for $t \in [0, 1]$. In addition, $\boldsymbol{x}_1 \sim p_{data}$ represents an observed sample, and $\boldsymbol{x}_0 \sim p_0$ a sample from a chosen base distribution. We further denote the conditional vector field as $u(\cdot, \cdot, \boldsymbol{z}) : [0, 1] \times \mathbb{R}^d \to \mathbb{R}^d$, equivalently as $u_t(\cdot, \boldsymbol{z}) : \mathbb{R}^d \to \mathbb{R}^d$ indexed by $t \in [0, 1]$. The path of conditional probability densities is denoted by $p_t(\cdot|\boldsymbol{z}) : \mathbb{R}^d \to \mathbb{R}^+$; and the conditional flow by $\phi_t(\cdot|\boldsymbol{z}) : \mathbb{R}^d \to \mathbb{R}^d, t \in [0, 1]$.

## 2.1 Continuous Normalizing Flow

Continuous normalizing flows (CNFs) describe probability distributions by the evolution of some probability density path. Denote the observed data by $\boldsymbol{x} \in \mathbb{R}^d$. Further, assume there exists a time-dependent vector field $v_t : \mathbb{R}^d \to \mathbb{R}^d$, $t \in [0, 1]$ that describes the evolution of a probability density path $p_t : \mathbb{R}^d \to \mathbb{R}^+$ indexed by $t \in [0, 1]$; we will use $v_t$ to describe the density of $\boldsymbol{x}$. The path then solves the continuity equation $\partial_t p_t = -\nabla \cdot (v_t p_t)$, which is the Fokker-Planck equation with zero diffusion. Due to the probabilistic representation theorem in Ambrosio et al. (2008, Theorem 8.2.1), the continuity equation admits a representation formulated as a solution of the ODE,

$$\frac{d}{dt}\phi_t(\boldsymbol{x}) = v_t(\phi_t(\boldsymbol{x})), \ \phi_0(\boldsymbol{x}) = \boldsymbol{x}_0, \tag{2}$$

where $\phi_t : \mathbb{R}^d \to \mathbb{R}^d$ is a push-forward map sending $\mu_0$ to $\mu_t = \phi_{t,\sharp}\mu_0$. This map is called *flow* in the machine learning literature (Chen et al., 2018; Grathwohl et al., 2018; Lipman et al., 2022). The log likelihood $f(t) = \log p_t(\phi_t(\boldsymbol{x}))$ at any point $\boldsymbol{x}$ can be obtained by solving the instantaneous change-of-variable formula forward in time, with initial conditions $c = \log p_0(\phi_0(\boldsymbol{x}))$, and the target $f(1) = \log p_1(\phi_1(\boldsymbol{x}))$:

$$\frac{d}{dt}\begin{pmatrix} \phi_t(\boldsymbol{x}) \\ f(t) \end{pmatrix} = \begin{pmatrix} v_t(\phi_t(\boldsymbol{x})) \\ -\nabla \cdot (v_t(\phi_t(\boldsymbol{x}))) \end{pmatrix}, \quad \begin{pmatrix} \phi_0(\boldsymbol{x}) \\ f(0) \end{pmatrix} = \begin{pmatrix} \boldsymbol{x}_0 \\ c \end{pmatrix}. \tag{3}$$

## 2.2 Flow Matching

Liu et al. (2022) and Lipman et al. (2022) concurrently introduced a similar training objective for learning flexible flow-based generative models, with NN parameterized vector field $v_t$,

$$\inf_\theta \mathbb{E}_{t, p_{data}(\boldsymbol{x}_1), p_t(\boldsymbol{x}_t|\boldsymbol{x}_1)} \|v_t(\boldsymbol{x}_t; \theta) - u_t(\boldsymbol{x}_t \mid \boldsymbol{x}_1)\|^2, \tag{4}$$

where $t \sim \mathcal{U}[0, 1]$, $\boldsymbol{x}_1 \sim p_{data}(\boldsymbol{x}_1)$, and now $\boldsymbol{x}_t \sim p_t(\boldsymbol{x}_t \mid \boldsymbol{x}_1)$. We refer to this objective as Flow Matching (FM). Flow matching resembles diffusion model with score matching except that the steps of noising (with conditional vector field $u_t(\boldsymbol{x}|\boldsymbol{x}_1)$) and denoising (with marginal vector field $v_t$) are deterministic. Solving Eq. 2 forward in time allows for generation from the learnt model.

The family of conditional vector field $u_t$ that governs the conditional probability path $p_t(\boldsymbol{x}_t|\boldsymbol{x}_1)$ is a design choice. Lipman et al. (2022) considered a particular example of the conditional probability path, $p_t(\boldsymbol{x}_t|\boldsymbol{x}_1) = N(\mu_t(\boldsymbol{x}_1), \sigma_t(\boldsymbol{x}_1)^2 I_d)$, where $\mu_t$ and $\sigma_t$ are time-dependent functions, with end points $\mu_0(\boldsymbol{x}_1) = 0$ and $\sigma_0^2(\boldsymbol{x}_1) = 1$ such that $p_0(\boldsymbol{x}_0|\boldsymbol{x}_1) \overset{d}{=} N(\boldsymbol{x}_0; 0, I_d)$. Therefore, the probability path $p_t = \varphi_t \sharp p_0$ is induced by the map $\varphi_t(\boldsymbol{x}) = \mu_t(\boldsymbol{x}_1) + \sigma_t(\boldsymbol{x}_1)\boldsymbol{x}$, which is the solution of the characteristic ODE $\frac{d}{dt}\varphi_t(\boldsymbol{x}) = u_t(\varphi_t(\boldsymbol{x})|\boldsymbol{x}_1)$. A special example includes linear interpolation in the Wasserstein space, $\varphi_t(\boldsymbol{x}) = (1 - t)\boldsymbol{x} + t\boldsymbol{x}_1$. For this choice, the corresponding conditional vector field is $u_t(\mathbf{x}_t|\mathbf{x}_1) = \frac{\mathbf{x}_1 - \mathbf{x}_t}{1-t}$ for $t \in [0, 1)$.

## 3 Structured Flow Autoencoders

In this section, we augment probabilistic graphical models with CNF likelihoods to design *structured flow autoencoders (SFAs)*, a family of structured flow-based probabilistic generative models.

**From marginal vector field to conditional vector field.** To enable probabilistic graphical modeling using flow-based models, we rely on a key insight arising from Bayes formula: the marginal vector field can be equivalently derived as the expectation of conditional vector field $v_t(\boldsymbol{x}|\boldsymbol{z})$ over an unobserved latent variable $\boldsymbol{z}$,

$$v_t(\boldsymbol{x}) = \int v_t(\boldsymbol{x}|\boldsymbol{z}) \frac{p_t(\boldsymbol{x}|\boldsymbol{z})p_t(\boldsymbol{z})}{\int p_t(\boldsymbol{x}|\boldsymbol{z})p_t(\boldsymbol{z})d\boldsymbol{z}} d\boldsymbol{z} = \mathbb{E}_{p_t(\boldsymbol{z}|\boldsymbol{x})}[v_t(\boldsymbol{x}|\boldsymbol{z})], \tag{5}$$

which also resembles the posterior predictive distribution. We formally state this result below in Theorem 3.1, which shows that $\mathbb{E}_{p_t(\boldsymbol{z}|\boldsymbol{x})}[v_t(\boldsymbol{x}|\boldsymbol{z})]$ is indeed the vector field that generates the path of marginal probability distributions $p_t(\boldsymbol{x})$. The proof is in Appendix A.1, which proceeds by verifying that Eq. 5 satisfies the continuity equation (Lipman et al., 2022).

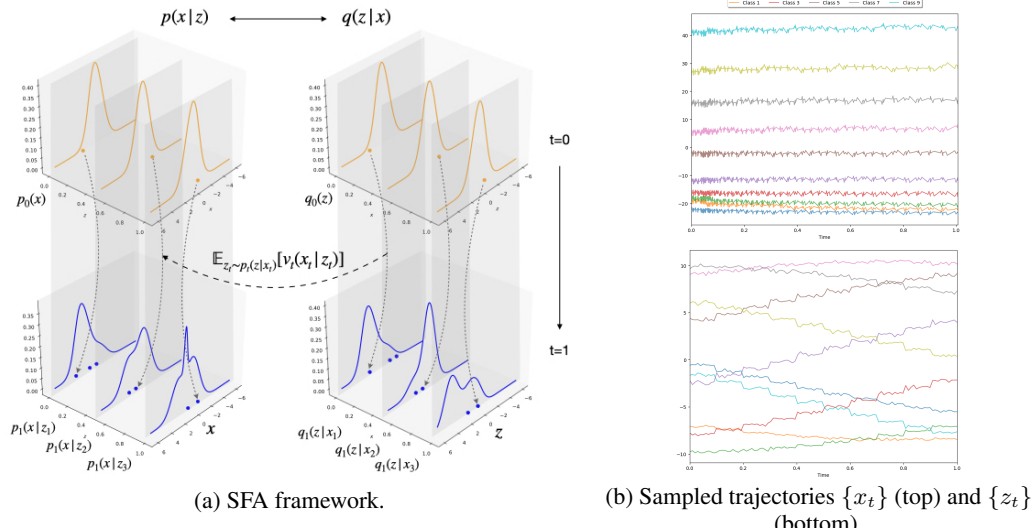

(a) SFA framework.

(b) Sampled trajectories $\{x_t\}$ (top) and $\{z_t\}$ (bottom).

Figure 3: Overview of Structured Flow Autoencoders (SFAs). **(a)** SFA framework showing co-evolving conditional probabilities for latent $\boldsymbol{z}$ and observed $\boldsymbol{x}$ with conditional CNFs. In the SCFM objective, we compute a convolution of conditional vector field $v_t$ for $\boldsymbol{x}_t(\cdot, \boldsymbol{z}_t)$ with respect to $q_t(\boldsymbol{z}_t|\boldsymbol{x}_t)$, when $\boldsymbol{x}_t = (1-t)\boldsymbol{x}_0 + t\boldsymbol{x}_1$. For conditional generation, sampling follows $\boldsymbol{z}_1 \sim p_1(\boldsymbol{z}_1), \boldsymbol{x}_1 \sim p_1(\boldsymbol{x}_1|\boldsymbol{z}_1)$; latent representation involves $\tilde{\boldsymbol{z}}_1 \sim q_1(\boldsymbol{z}_1|\boldsymbol{x}_1)$. **(b)** 1D PCA projected sampled trajectories on MNIST. **Top**: observation paths given the posterior sample $z_1$. **Bottom**: latent paths given observed $x_1$ sampled from different digit classes. The latent trajectories diverge smoothly from shared noise, showing SFA learns structured transport; the likelihood paths conditioned on $z_1$ are cleanly separated by digit classes, confirming the posterior encodes meaningful latent structure.

**Theorem 3.1.** *Given conditional vector field $v_t(\boldsymbol{x}|\boldsymbol{z})$ that generates the path $\{p_t(\boldsymbol{x}|\boldsymbol{z})\}$ of probability kernel for $p(\boldsymbol{z})$ a.e. $\boldsymbol{z}$. Then $v_t(\boldsymbol{x}) = \mathbb{E}_{p_t(\boldsymbol{z}|\boldsymbol{x})}[v_t(\boldsymbol{x}|\boldsymbol{z})]$ is the marginal vector field that generates the marginal probability path $p_t(\boldsymbol{x}) = \int p_t(\boldsymbol{x}|\boldsymbol{z})\,p(\boldsymbol{z})\,d\boldsymbol{z}$ over $t \in [0,1]$ under regularity conditions. Conversely, the marginal probability path $p_t(\boldsymbol{x})$ induced by $v_t(\boldsymbol{x})$ via the continuity equation coincides with $\int p_t(\boldsymbol{x}|\boldsymbol{z})\,p(\boldsymbol{z})\,d\boldsymbol{z}$.*

Theorem 3.1 allows us to gain flexibility and interpretability in flow-based generative modeling by introducing latent structure to the otherwise marginal vector field of data distribution. This realization is the key to uncovering rich latent structure while ensuring marginal distribution is captured faithfully.

**Structured Flow Autoencoders (SFA).** This motivates us to design structured flow autoencoders (SFAs). This family of probabilistic autoencoders consists of co-evolving probability paths $\{p_t(\cdot|\boldsymbol{z}_t)\}$ and $p_t(\cdot|\boldsymbol{x}_t)$ across time $t \in [0,1]$; and these paths are connected to the observed data distribution $p_t(\boldsymbol{x}_t)$ through $\mathbb{E}_{p_t(\boldsymbol{z}|\boldsymbol{x})}[v_t(\boldsymbol{x}|\boldsymbol{z})]$ (Figure 3a). At $t = 1$, the probability $p_1(\cdot|\boldsymbol{x}_1)$ and $p_1(\cdot|\boldsymbol{z}_1)$ corresponds to the model likelihood and posterior for the observed data; at $t = 0$, the probabilities correspond to the marginal base distributions that are easy to sample and evaluate. In addition, Theorem 3.1 allows the specification of any posterior family, which could have inbuilt structure according to a graphical model. We defer three representative examples to the next subsections.

**Structured Conditional Flow Matching (SCFM).** To learn SFAs, we match the marginal path $p_t(\boldsymbol{x}_t)$ induced from $\mathbb{E}_{p_t(\boldsymbol{z}|\boldsymbol{x})}[v_t(\boldsymbol{x}|\boldsymbol{z})]$ to a preselected path as in FM objective, and optimize for the conditional vector field $v_t$ and posterior approximation $q \in Q = \{(t, x) \mapsto q_t(\boldsymbol{z}|\boldsymbol{x}), (t, x) \in [0,1] \times \mathcal{X}\}$. We formalize this approach through the *Structured Conditional Flow Matching (SCFM)* objective:

$$\mathcal{R}(\theta, q) = \mathbb{E}_{\substack{\boldsymbol{x}_1 \sim p_{data}(\boldsymbol{x}_1) \\ \boldsymbol{x}_t \sim p_t(\boldsymbol{x}|\boldsymbol{x}_1),\ t \sim Unif[0,1]}} \left\| \mathbb{E}_{q_t(\boldsymbol{z}_t|\boldsymbol{x}_t)}[v_t(\boldsymbol{x}_t|\boldsymbol{z}_t; \theta)] - u_t\left(\boldsymbol{x}_t \mid \boldsymbol{x}_1\right) \right\|^2, \qquad (6)$$

where the outer expectation is w.r.t. the conditional flow trajectory given observed samples $\boldsymbol{x}_1$; the inner expectation is w.r.t. the trajectory of the posterior distribution; the reference vector field $u_t$ is

chosen *a priori*, which defines the desired trajectory connecting observed data to the base distribution $p_0$. Intuitively, SCFM solves "de-mixing" problem, decomposing the observed signal to (1) the data generation model and (2) the latent structure components.

**Posterior approximation.** The choice of approximating family $Q$ must be sufficiently expressive to capture the complexity of the true posterior, while not so complex as to destabilize the training. We discuss specific choices for each latent structure in Figure 2 in the following subsection. Now, with a learned posterior approximation, the corresponding marginal distribution (prior) in the latent space can be derived post-hoc. Motivated by empirical Bayes, this can be achieved via integration over the observation marginal: $q_t(\boldsymbol{z}_t) = \int q_t(\boldsymbol{z}_t|\boldsymbol{x}_t)p(\boldsymbol{x}_t)d\boldsymbol{x}_t$. In practice, a separate model can be used to learn the marginal after training (Wang et al., 2023; Preechakul et al., 2022).

**VAE vs SFA.** SFA shares the motivation of structured VAEs for learning probabilistic encoders and decoders, but replaces the likelihood-based ELBO with an objective grounded in the consistency of marginal probability paths (Theorem 3.1). First, while CNFs can parameterize VAE encoders and decoders, doing so requires expensive likelihood evaluation at every training step; SFA avoids this entirely. Second, $\beta$-VAE (Higgins et al., 2017) introduces a regularization parameter to balance the reconstruction and KL terms, trading off density estimation quality for $p(\mathbf{x}|\mathbf{z})$ against $p(\mathbf{z}|\mathbf{x})$. SFA requires no such trade-off: the latent $\mathbf{z}$ captures meaningful structure in the conditional path $p_t(\mathbf{x}_t|\mathbf{z}_1)$, which simultaneously stabilizes training and ensures accurate marginal reconstruction. Figure 3b provides a visualization on samples of $z_t$ and $x_t$ trajectory from MNIST dataset.

For different graphical models (cf. Figure 2), SCFM objective can be adapted to accommodate their specific structures. We illustrate through three examples in the following subsections, spanning continuous, finite mixture and Markov dynamic latent structure. We chose these three examples because they are (1) widely applicable across different domains, (2) representative of different dependency types (continuous, finite mixture, temporal), and (3) sufficient to demonstrate the framework's flexibility.

## 3.1 CONTINUOUS LATENT VARIABLE MODEL

Consider the graphical model in Figure 2a, where $\boldsymbol{z} \in \mathbb{R}^p$ and $\boldsymbol{x} \in \mathbb{R}^d$, $0 < p < d$,

$$\boldsymbol{z}_i \overset{iid}{\sim} p(\boldsymbol{z}), \quad \boldsymbol{x}_i|\boldsymbol{z}_i \overset{ind}{\sim} p(\boldsymbol{x}|\boldsymbol{z}),$$

giving rise to the posterior $\boldsymbol{z}_i|\boldsymbol{x}_i \sim p(\boldsymbol{z}|\boldsymbol{x})$. We estimate both the unknown likelihood and posterior from observed data, $\boldsymbol{x}_{1,i} \sim p_{data}(\boldsymbol{x}) = \int p(\boldsymbol{x}|\boldsymbol{z})p(\boldsymbol{z})d\boldsymbol{z}$, under SCFM objective. Following from Theorem 3.1, the likelihood model is the conditional CNF generated by the conditional vector field $v_t(\boldsymbol{x}_t|\boldsymbol{z}_t, \theta)$:

$$\frac{d}{dt}\phi_t(\boldsymbol{x}) = v_t(\phi_t(\boldsymbol{x})|\boldsymbol{z}; \theta), \quad \phi_0(\boldsymbol{x}) = \boldsymbol{x}_0, \quad \boldsymbol{x}_0 \sim p_0(\boldsymbol{x}). \tag{7}$$

In this example, the risk function follows directly from Eq. 6. For practical implementation, the mixed conditional vector field $\mathbb{E}_{q_t(\boldsymbol{z}_t|\boldsymbol{x}_t)}[v_t(\boldsymbol{x}_t|\boldsymbol{z}_t)]$ can be approximated with a single sample $\tilde{\boldsymbol{z}} \sim q_t(\boldsymbol{z}|\boldsymbol{x})$, as commonly used in VAE (Kingma & Welling, 2013). The approximation family $Q$ for the posterior path $\{q_t(\mathbf{z} \mid \mathbf{x})\}_{t \in [0,1]}$ admits different choices, each defining a valid absolutely continuous curve in $(\mathcal{P}_2(\mathbb{R}^p), W_2)$.

**Conditional CNF.** One can parameterize $q_t(z_t|x_t)$ via a conditional vector field $r_t(\mathbf{z} \mid \mathbf{x}; \theta)$, defining the posterior path as the pushforward of a base distribution $q_0(\mathbf{z})$ through the flow ODE $s \in [0, t]$:

$$\frac{d}{ds}\psi_s(\mathbf{z}) = r_s(\psi_s(\mathbf{z}) \mid \mathbf{x}_t; \theta), \quad \psi_0(\mathbf{z}) = \mathbf{z}_0, \quad \mathbf{z}_0 \sim q_0(\mathbf{z}).$$

This is maximally expressive. The path can traverse all of $\mathcal{P}_2(\mathbb{R}^p)$, but it requires backpropagation through adjoint ODE steps (Chen et al., 2018) to evaluate the inner expectation, adding computational cost and potential training instability.

**Gaussian family.** Alternatively, $Q$ can be a parametric family indexed by $t$ and $\mathbf{x}$ : $Q = \{(t, \mathbf{x}) \mapsto N(\mu_\theta(t, \mathbf{x}), \Sigma_\theta(t, \mathbf{x}))\}$, which parameterizes a path in the Bures-Wasserstein submanifold. Gradients follow directly from the reparameterization trick, offering computational efficiency and training stability.

## 3.2 LATENT FINITE MIXTURE MODEL

In this section, we consider the generative model in Figure 2b, where the latent variable $z$ follows a finite mixture distribution with $K$ classes. This graphical model takes into account the latent class $\xi \in [K]$, where $p(\xi_i = k|\pi) = \pi_k$ for each sample $x$. It gives rise to posteriors on the local class label $\xi_i$, the continuous latent $z$, and the global class proportion $\pi$.

**Generative Model**

$$\pi \sim p(\pi), \; \xi_i|\pi \overset{iid}{\sim} \text{Cat}(\pi),$$

$$z_i|\xi_i \sim p(z|\xi_i), \; x_i|z_i \overset{ind}{\sim} p(x|z_i),$$

**Inference Model**

$$\xi_i|x_i, \pi \sim \text{Cat}(p(\xi_i|x_i, \pi)),$$

$$z_i|x_i, \; \xi_i \sim p(z_i|x_i, \xi_i), \pi|z_{[n]} \sim p(\pi|z_{[n]}).$$

When $\pi \sim Dir(\alpha)$, the posterior for overall proportions $p(\pi|z_{[n]})$ has a closed form $Dir(\tilde{\alpha})$ with $\tilde{\alpha}_k = \alpha_k + \sum_{i=1}^{n} \mathbf{1}\{\xi_i = k\}$. The local label $\xi$, $z$ are of major interest for drawing inference on the latent class assignment and value. Next, we adapt SCFM for latent finite mixture model: both $\xi$ and $z$ are now integrated out in the inner expectation. Applying Theorem 3.1 gives rise to Eq. 8.

$$\inf_{q \in Q, \theta \in \Theta} \mathbb{E}_{\substack{x_1 \sim p_{data}(x_1) \\ x \sim p_t(x|x_1) \\ t \sim Unif[0,1]}} \left\| \mathbb{E}_{q_t(\xi_t|x_t)q_t(z_t|x_t,\xi_t)}[v_t(x_t|z_t; \theta)] - u_t(x_t \mid x_1) \right\|^2. \tag{8}$$

The design of likelihood model follows similarly as in Eq. 7, which is a CNF conditioned on $z$ only. The approximation family for $p_t(\xi_i|x_i)$ could be chosen as a Gumbel-Softmax distribution with time-dependent parameters, alternatively constant across $t$ to reduce the complexity of the model. The approximation family for $p_t(z|x, \xi)$, should be chosen as conditional CNF or parametric distribution indexed by $t, x, \xi$. As $z$ is unconstrained, a Gaussian approximation family can be posited similarly as in Section 3.1.

## 3.3 LATENT DYNAMIC SYSTEM

We consider discrete-time sequential generation with continuous latent states following the graphical model in Figure 2c. For index $s \in [S]$, the generative model is formalized as conditionally independent observations $x^s$ given latent state $z^s$; the inference model is focused on the full posterior of latent trajectory $z^{[S]}$ given the observation sequence $x^{[S]}$, which can be factorized into full conditionals at each index $s$ given all previous history.

**Generative Model** $\quad z_i^s|z_i^{s-1} \sim p(z|z^{s-1}), \quad x_i^s|z_i^s \sim p(x^s|z_i^s).$

**Inference Model** $\quad z_i^{[S]}|x_i^{[S]} \sim p(z^{[S]}|x^{[S]}) = \prod_{s \in [S]} p(z^s|z^{[s-1]}, x^{[S]}).$

Given observed sample sequences $\{x_i^{[S]}\}_{i=1}^n$, the SCFM objective can be shown to have the form in Eq. 9. Accompanying theoretical results and numerical derivations are detailed in Theorem A.2, extending Theorem 3.1.

$$\inf_{q \in Q, \theta \in \Theta} \mathbb{E}_{\substack{x_1 \sim p_{data}, x_t \sim p_t(x_t|x_1) \\ t \sim \text{Unif}[0,1]}} \left\| \sum_{s \in [S]} \mathbb{E}_{p_t(z_t^{[S]}|x_t^{[S]})}[v_t(x_t^s|z_t^s; \theta)] - u_t(x_t^s \mid x_1^s) \right\|^2. \tag{9}$$

Here the sum over $s \in [S]$ captures the sequential dependencies inherent in the model structure. Following the assumed conditional independence, we parameterize the likelihood using conditional CNFs for each $x^s, s \in [S]$ according to Eq. 7. To approximate the posterior $p_t(z^{[S]}|x^{[S]})$, we employ a parametric family with its parameters indexed by $t \in [0, 1]$, previous states $z^{[s-1]}$, and the full observation sequence $x^{[S]}$. Details of implementations can be found in Appendix B.4.

## 4 EMPIRICAL STUDIES

In this section, we evaluate the proposed SFA across a range of tasks and data modalities: (a) conditional density estimation for Pinwheel dataset; (b) latent clustering on MNIST dataset; (c) gene expression modelling on single-cell RNA-seq data; (d) sequence modelling with Pendulum trajectory

Table 2: Comparison of metrics for MNIST dataset between VAE, latent FM and SFA. Evaluated on a held-out set of size 1000. The OOD dataset consists of first 10 classes of letters and the first 10 classes of digits in EMNIST. The clustering is done in the latent space via k-means with $k$ given.

| | $\log p(x\|z)\uparrow$ | $\log p(z\|x)\uparrow$ | Vendi $\uparrow$ | SSIM $\uparrow$ | NMI (OOD) $\uparrow$ | ARI(OOD) $\uparrow$ |
|---|---|---|---|---|---|---|
| VAE | $-\mathbf{453.648}$ | $-85.448$ | $\mathbf{63.286}$ | 0.419 | 0.039(0.033) | 0.017(0.012) |
| VampVAE | $-584.845$ | 155.820 | 1.140 | 0.866 | 0.006(0.006) | 0.000(0.000) |
| Latent FM | - | - | 8.380 | $\mathbf{0.980}$ | 0.488(0.392) | $\mathbf{0.381}$(0.194) |
| Latent FM (VAE) | $-910.925$ | $-11.192$ | 19.631 | 0.697 | 0.309(0.152) | 0.205(0.073) |
| SFA | $-916.901$ | $\mathbf{793.262}$ | 25.589 | 0.716 | 0.490($\mathbf{0.394}$) | 0.356($\mathbf{0.208}$) |
| w/Deterministic | $-858.385$ | - | 10.189 | 0.732 | $\mathbf{0.501}$(0.333) | 0.379(0.155) |
| w/CNF Posterior | $-907.998$ | 356.141 | 23.166 | 0.654 | 0.485(0.325) | 0.355(0.118) |

video dataset. We compare our method mainly to VAE counterparts, including SVAE (Johnson et al., 2016), VampVAE (Tomczak & Welling, 2018), $\beta$-VAE (Higgins et al., 2017), Gaussian Mixture VAE (GMVAE) (Lin et al., 2018); Latent Flow Matching methods with deterministic autoencoders (LatentFM) (Dao et al., 2023) and probabilistic autoencoders (LatentFM w/VAE) that are trained with two stages: first stage with encoder and decoders only then jointly with latent FM in the second stage. In the following, we use Mixture-SVAE to refer to the SVAE with Gaussian mixture latent, and equivalently GMVAE; we use GLD-SVAE to refer to the Gaussian Latent dynamics SVAE.

For a fair comparison, we restrict the prior to be fixed in SVAE and only focus on modeling the conditional probabilities. All the metrics are evaluated based on samples held out from training. More experimental details and supporting visualizations are hosted in Appendix B. We draw comparisons in the quality of (1) generated posterior samples (or latent distribution samples for Latent FM) and (2) generated posterior predictive samples, and downstream tasks, such as latent space clustering.

For (1), given a large training dataset, the posterior distribution should be able to provide an accurate representation of the latent distribution. Therefore, under simulation settings where latent ground truth is known, we are able to evaluate the discrepancy between learned latent representation and the truth, provided $z \sim q_1(z|x)$ with $x \sim p_{data}(x)$.

For (2), we conduct posterior predictive check to evaluate the discrepancy of the samples from $p_{data}(x)$ and $p_{pred}(\tilde{x}|x) = \int p_1(\tilde{x}|z)p(z|x)dz$. To sample from the latter, we follow

$$x \sim p_{data}(x), \quad z|x \quad \sim q_1(z|x), \quad \tilde{x}|z \sim p_1(x|z).$$

We evaluate the diversity of the generated samples using Vendi score (Friedman & Dieng, 2022); quality of image generation with SSIM (Wang et al., 2004); quality of latent clustering with ARI (Hubert & Arabie, 1985), NMI (Strehl & Ghosh, 2002) and probabilistic version softNMI (Eq. 16).

**Summary of findings.** In conditional density modeling (Pinwheel), SFA consistently outperforms LatentFM and VAE-based models, showing better data density reconstruction and better latent space modelling. To assess scalability, we apply SFA to a single-cell RNA-seq dataset, where it effectively models high-dimensional gene expression data and outperforms VAEs in reconstruction quality. On image data (MNIST), both SFA and its mixture extension (Mixture-SFA) learn meaningful latent representations, generate high-fidelity samples, and perform well on latent-space clustering tasks. Finally, we highlight SFA's versatility on sequential data using the pendulum video dataset, where it successfully captures the low-dimensional periodic structure of the underlying physical system.

**Comparing posterior family.** When the latent is lower-dimensional, a smaller posterior model is sufficient relative to the model size needed for the likelihood. When learning multiple components jointly, simpler parametric approximation families are preferred over conditional CNFs for training stability. Computation-wise, SFA (2.4M parameters) requires $13.220 \pm 1.848$ seconds per epoch, comparable to VAE's $12.789 \pm 2.011$ seconds. With CNF as the posterior of SFA, we observe a drastic increase in the runtime to $167.460 \pm 176.817$ seconds. This is due to the additional time arising from sampling CNF during training, which requires solving an ODE at each gradient evaluation step.

Table 3: Subspace clustering on MNIST with latent mixtures models Mixture-SVAE and Mixture-SFA. Evaluated on a held-out set of size 1000.

|  | $\log p(x|z) \uparrow$ | $\log p(z|x) \uparrow$ | SSIM $\uparrow$ | softNMI $\uparrow$ | NMI $\uparrow$ | ARI $\uparrow$ |
|---|---|---|---|---|---|---|
| Mixture-SVAE | $-1133$ | 0.667 | 0.634 | 0.698 | 0.161 | 0.072 |
| Mixture-SFA | $\mathbf{-906}$ | $\mathbf{725}$ | $\mathbf{0.779}$ | $\mathbf{0.728}$ | $\mathbf{0.489}$ | $\mathbf{0.332}$ |

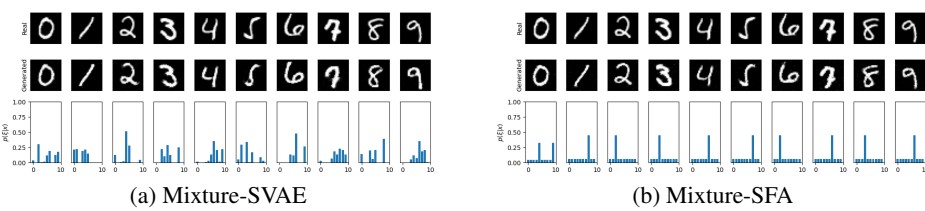

(a) Mixture-SVAE  (b) Mixture-SFA

Figure 4: Comparison of Mixture-SVAE and Mixture-SFA on MNIST dataset. The first row displays the posterior predictive $x_i \sim \int p_1(x|z)q_1(z|x_{1,i})dz$ and latent class assignment probability $\xi_i \sim q_1(\xi|x_{1,i})$ for a test data $x_{1,i}$. The second row shows the latent representation learned for digits, where each point is sampled from $z_i \sim q_1(z|x_{1,i})$ for a test data sample $x_{1,i}$.

### 4.1 MIXTURE MODELING: PINWHEEL DATA

We first illustrate the ability of SFA in learning conditional distributions using the toy example of the pinwheel dataset, with five clusters each having the shape of a blade (Johnson et al., 2016). The class membership is not provided during training. The goal is to evaluate if the posterior is able to uncover the latent structure of the data, and whether the model is able to capture the observed data distribution. In Figure 1, we visualize the generated data together with their representation coded in 1D colorbar. SFA is able to reconstruct the support of the ground truth distribution, in addition to capturing a meaningful latent representation for the angular rotation. In contrast, both the Latent FM and VAE-estimated density do not have well-separated components. As shown in Table 1, SFA based methods achieve similar density estimation quality as FM and comparable to ground truth, while SVAE based methods fail to model the density accurately.

### 4.2 IMAGE MODELING: MNIST DATA

Next, consider MNIST dataset LeCun et al. (2010). We aim to recover the probabilistic assignment of each image to the 10 classes, and learn a low dimensional feature representation at the same time. We first consider the graphical model with continuous latent, and compare SFA to VAE and Latent FM on the latent space clustering task. In addition, to check if the learned latent space captures desirable structures in the data, such as stroke and abstract shape, we sample from the latent distribution encoding Out-of-Distribution (OOD) data in the EMNIST dataset Cohen et al. (2017). Results and comparisons are summarized in Table 2. Visualizations are presented in Figure 6. Owing to posterior collapse, VAE learns an unstructured latent space with close to random samples (high Vendi score). SFA achieves the best balance of diversity (Vendi=25.6) and clustering (NMI=0.490) among all methods, while Latent FM trades off diversity for reconstruction (Vendi=8.4 but SSIM=0.980). Performance using mixture graphical model are organized in Table 3 and Figure 4. Mixture-SVAE improves generation quality and latent class separation over VAE. However, Mixture-SFA still achieves better clustering quality.

### 4.3 GENE EXPRESSION MODELING: SINGLE-CELL RNA-SEQ DATA

The dataset obtained from Lotfollahi et al. (2023) includes PBMCs from eight patients with Lupus. The data consists of 7 cell types, and treated and control with IFN-$\beta$ (Kang et al., 2018). The observed count is normalized and $\log(x + 1)$ transformed, then $5,000$ HVGs are selected. We apply continuous latent to learn the low-dim representation of the high-dimensional differential expression data. Figure 5 shows that both SFA and Latent FM produce meaningful clusters of the cell type in the latent space. SFA matches Latent FM in clustering accuracy while producing substantially more diverse samples, as indicated by the higher Vendi score and clustering scores (Table 4a).

Table 4: Comparison of metrics across different datasets and methods. (a) Kang HVG dataset evaluated on a held-out set of size 500. The observation has dimension 5000, due to the size, the log likelihood for CNF cannot be directly computed by solving adjoint-ODE, therefore left out of the comparison. (b) Pendulum dataset over posterior samples of observed ($\text{RMSE}_x$), and latent ($\text{RMSE}_z$). Evaluated on a held-out set of size 300.

(a) HVG Single-Cell RNA-Seq

|  | $\log p(z\|x)\uparrow$ | Vendi $(\boldsymbol{x})\uparrow$ | NMI $\uparrow$ | ARI $\uparrow$ |
|---|---|---|---|---|
| VAE | $-40.04$ | 26.58 | 0.412 | 0.257 |
| LatentFM | - | 5.801 | 0.617 | 0.457 |
| SFA | **384.1** | **737.7** | **0.633** | **0.460** |

(b) Pendulum Trajectory

|  | $\text{RMSE}_x\downarrow$ | $\text{RMSE}_z\downarrow$ |
|---|---|---|
| GLD-SVAE | 4.574 | 8.090 |
| LDS-SFA | **3.233** | **1.526** |

## 4.4 SEQUENTIAL MODELING: PENDULUM TRAJECTORY VIDEO DATA

We choose a pendulum trajectory dataset for LDS example. The dynamics is driven by the pendulum physical system modeled as a damped harmonic oscillator. The latent trajectory is 2 dimensional, consisting of angle and angular velocity. The observation is a video with discrete time frames mapped from latent trajectory. We compare SVAE with SFA in Table 4b, where we measure the discrepancy between the generated and ground truth of both observed and latent dynamics using RMSE. LDS-SFA reduces latent RMSE by over $5\times$ compared to GLD-SVAE (1.526 vs 8.090), suggesting better recovery of the underlying dynamics. Additional details on model implementations are in Appendix B.4.

## 5 DISCUSSION

In this work, we introduced structured flow autoencoders (SFA), a framework that integrates flow matching (FM) with latent probabilistic graphical models (PGMs) to achieve both high-fidelity generation and structured latent representation learning, leveraging the consistency of marginal probability paths (Theorem 3.1). At the core of SFA is the structured conditional flow matching (SCFM) objective, which extends flow matching by explicitly modeling latent variables, enabling joint estimation of the generative likelihood $p(\boldsymbol{x}|\boldsymbol{z})$ and the posterior $p(\boldsymbol{z}|\boldsymbol{x})$. A key contribution of this work is the generality of the SFA framework: SCFM provides a principled recipe for augmenting any graphical model that specifies conditional independence structure with flow-matching-based likelihoods. We instantiated this for continuous, finite mixture, and dynamical latent structures, and demonstrated consistent improvements over VAE-based and latent flow matching baselines across synthetic density estimation, image clustering, high-dimensional gene expression modeling, and sequential video generation in both generation quality and representation utility.

**Limitations and future work.** SFA with parametric posteriors is computationally comparable to VAEs, but the CNF posterior variant remains expensive due to ODE solves at each gradient step. In practice, simpler families suffice when the latent dimension is much smaller than the observation dimension; principled selection strategies remain an open question. In addition, our contribution is primarily methodological: formulating how to compose representation learning with flow matching via SCFM. Scaling to complex image datasets poses a separate, architectural challenge: designing deep neural net decoders that meaningfully condition on a simultaneously learned stochastic latent, rather than bypassing it through skip connections. Addressing this architectural challenge is a natural next step that complements the methodological framework introduced here. On the methodological side, a parallel direction is to extend SFA toward learning identifiable, decomposable latents that can be manipulated for intervention and out-of-domain generation.

## ACKNOWLEDGMENTS

This work was supported in part by funding from the Office of Naval Research under grant N00014-23-1-2590, the National Science Foundation under grant No. 2310831, No. 2428059, No. 2435696, No. 2440954, a Michigan Institute for Data Science Propelling Original Data Science (PODS) grant, Two Sigma Investments LP, and LG Management Development Institute AI Research. Any opinions, findings, and conclusions or recommendations expressed in this material are those of the authors and do not necessarily reflect the views of the sponsors.

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

## A    THEORETICAL DETAILS

We formally present the probabilistic representation of solutions to the continuity equation when the vector field fails to be Lipschitz w.r.t $x$. In this case, the solution to the characteristic ODE (flow ODE) is not unique. When using neural nets to parameterize the vector field, we want to verify that the solution of the ODE indeed induces a solution to the continuity equation.

Firstly, we denote $\mu_t : [0,1] \to \mathcal{P}(\mathbb{R}^d)$ as the path of probability indexed by $t$, $AC^p(0,1;\mathbb{R}^d)$ as the space of absolutely continuous curves $\gamma : [0,1] \to \mathbb{R}^d$ with finite $p$ energy, i.e. $|\gamma'| \in L^p(0,1)$. Denote $\Gamma$ as the space of continuous map $\gamma : [0,1] \to \mathbb{R}^d$. Let $e_t : (x,\gamma) \mapsto \gamma(t)$ as the evaluation map. Then define the curve of probability measure induced by the evaluation map as

$$\mu_t^\eta = e_t \sharp \eta, \ t \in [0,1]$$

where by definition,

$$\int \psi(x) d\mu_t^\eta(x) = \int_{\mathbb{R}^d \times \Gamma} \psi(\gamma(t)) d\eta(x,\gamma), \ \forall \psi \in C_b^0(\mathbb{R}^d), \ t \in [0,1].$$

Then finally recall that the continuity equation

$$\partial_t \mu_t + \nabla \cdot (v t \mu_t) = 0 \quad \text{in } \mathbb{R}^d \times (0,1).$$

**Theorem A.1** (Ambrosio et al. (2008) Theorem 8.2.1). *Let $\mu_t : [0,1] \to \mathcal{P}(\mathbb{R}^d)$ be a narrowly continuous solution of the continuity equation for a suitable Borel vector field $v_t(x)$ such that for some $p > 1$,*

$$\int_0^1 \int_{\mathbb{R}^d} |v_t(x)|^p d\mu_t(x) dt < +\infty.$$

*i Then (a) there exists a probability measure $\eta$ in $\mathbb{R}^d \times \Gamma$ such that that concentrates on the set of pairs $(x,\gamma)$ such that $\gamma \in AC^p(0,1;\mathbb{R}^d)$ is a solution of the ODE $\dot{\gamma}(t) = v_t(\gamma(t))$ for $L^1$-a.e. $t \in [0,1]$ with $\gamma(0) = x$. and (b) $\mu_t = \mu_t^\eta \ \forall t \in [0,1]$.*

*ii Conversely, any $\eta$ satisfies (a) and $\int_0^1 \int_{\mathbb{R}^d \times \Gamma} |v_t(\gamma(t))| d\eta(x,\gamma) dt < +\infty$ induces a solution of the continuity equation via $\mu_t^\eta = e_t \sharp \eta$, with $\mu_0 = e_0 \sharp \eta$.*

The converse argument can be readily extended to the conditional vector field $v(\cdot,\cdot,z) : [0,1] \times \mathbb{R}^d \to \mathbb{R}^d$ for any $\nu$-a.e. $z$. Then the solution curve to the characteristic ODE would be indexed by $z$, denoted as $\gamma_z$.

### A.1    PROOFS IN § 3

In the following, we use the same notation in the main paper for the proof details in § 3. Recall the marginal vector field defined via the posterior expectation (Eq. 5):

$$v_t(x) = \int v_t(x|z) \frac{p_t(x|z) \, p(z)}{\int p_t(x|z') \, p(z') \, dz'} \, dz = \mathbb{E}_{p_t(z|x)}[v_t(x|z)].$$

The correct notion of distance between velocity fields, consistent with the Wasserstein-2 geometry of Ambrosio et al. (2008) (Chapter 8), is the *kinetic energy norm*:

$$\|w\|_{L^2(p_t \otimes \mathcal{L}^1)}^2 = \int_0^1 \int_{\mathbb{R}^d} |w_t(x)|^2 \, p_t(x) \, dx \, dt,$$

which appears as the action functional in the Benamou–Brenier formula for $W_2$.

We now make precise the sense in which the conditional vector fields $\{v_t(\cdot|z)\}_z$ give rise to a *consistent* marginal dynamics, restating the theorem for completeness.

**Theorem** (Theorem 3.1). *Given conditional vector field $v_t(x|z)$ that generates the path $\{p_t(x|z)\}$ of probability kernel for $p(z)$ a.e. $z$. Then $v_t(x) = \mathbb{E}_{p_t(z|x)}[v_t(x|z)]$ is the marginal vector field that generates the marginal probability path $p_t(x) = \int p_t(x|z) \, p(z) \, dz$ over $t \in [0,1]$ under regularity conditions. Conversely, the marginal probability path $p_t(x)$ induced by $v_t(x)$ via the continuity equation coincides with $\int p_t(x|z) \, p(z) \, dz$.*

*Proof.* The proof proceeds in two steps: (i) the mixed vector field solves the marginal continuity equation, and (ii) uniqueness of the induced probability path.

**Regularity conditions.** Assume:

(R1) $v_t(\boldsymbol{x}|\boldsymbol{z})$ is Borel measurable in $(\boldsymbol{x}, \boldsymbol{z})$ and continuous in $t$, with $\int_0^1 \int |v_t(\boldsymbol{x}|\boldsymbol{z})|^2 \, p_t(\boldsymbol{x}|\boldsymbol{z}) \, d\boldsymbol{x} \, dt < \infty$ for $p(\boldsymbol{z})$-a.e. $\boldsymbol{z}$.

(R2) $v_t(\boldsymbol{x}|\boldsymbol{z}) \, p_t(\boldsymbol{x}|\boldsymbol{z})$ and $\nabla_{\boldsymbol{x}} \cdot \big(v_t(\boldsymbol{x}|\boldsymbol{z}) \, p_t(\boldsymbol{x}|\boldsymbol{z})\big)$ are continuous and bounded in $(\boldsymbol{x}, t, \boldsymbol{z})$.

(R3) $p_t(\boldsymbol{x}|\boldsymbol{z})$ and $\frac{d}{dt} p_t(\boldsymbol{x}|\boldsymbol{z})$ are continuous in $(t, \boldsymbol{z})$, and $\frac{d}{dt} p_t(\boldsymbol{x}|\boldsymbol{z})$ is uniformly bounded for all $t \in [0, 1]$ and $p(\boldsymbol{z})$-a.e. $\boldsymbol{z}$.

These conditions ensure the Leibniz integral rule applies, justifying the exchange of $\frac{d}{dt}$ and $\nabla_{\boldsymbol{x}} \cdot$ with $\int \cdot \, p(\boldsymbol{z}) \, d\boldsymbol{z}$.

By assumption, for $p(\boldsymbol{z})$-a.e. $\boldsymbol{z}$, the conditional pair $(v_t(\cdot|\boldsymbol{z}), \, p_t(\cdot|\boldsymbol{z}))$ satisfies

$$\frac{d}{dt} p_t(\boldsymbol{x}|\boldsymbol{z}) = -\nabla_{\boldsymbol{x}} \cdot \big(v_t(\boldsymbol{x}|\boldsymbol{z}) \, p_t(\boldsymbol{x}|\boldsymbol{z})\big). \tag{10}$$

Integrating against $p(\boldsymbol{z})$ and applying Leibniz (justified by (R2)–(R3)):

$$\begin{aligned}
\frac{d}{dt} p_t(\boldsymbol{x}) &= \int \frac{d}{dt} p_t(\boldsymbol{x}|\boldsymbol{z}) \, p(\boldsymbol{z}) \, d\boldsymbol{z} \\
&= -\int \nabla_{\boldsymbol{x}} \cdot \big(v_t(\boldsymbol{x}|\boldsymbol{z}) \, p_t(\boldsymbol{x}|\boldsymbol{z})\big) \, p(\boldsymbol{z}) \, d\boldsymbol{z} \\
&= -\nabla_{\boldsymbol{x}} \cdot \int v_t(\boldsymbol{x}|\boldsymbol{z}) \, p_t(\boldsymbol{x}|\boldsymbol{z}) \, p(\boldsymbol{z}) \, d\boldsymbol{z}.
\end{aligned} \tag{11}$$

Now we rewrite the integrand by multiplying and dividing by $p_t(\boldsymbol{x})$:

$$\begin{aligned}
\int v_t(\boldsymbol{x}|\boldsymbol{z}) \, p_t(\boldsymbol{x}|\boldsymbol{z}) \, p(\boldsymbol{z}) \, d\boldsymbol{z} &= \int v_t(\boldsymbol{x}|\boldsymbol{z}) \frac{p_t(\boldsymbol{x}|\boldsymbol{z}) \, p(\boldsymbol{z})}{p_t(\boldsymbol{x})} \, p_t(\boldsymbol{x}) \, d\boldsymbol{z} \\
&= \left(\int v_t(\boldsymbol{x}|\boldsymbol{z}) \, p_t(\boldsymbol{z}|\boldsymbol{x}) \, d\boldsymbol{z}\right) p_t(\boldsymbol{x}) \\
&= \mathbb{E}_{p_t(\boldsymbol{z}|\boldsymbol{x})}[v_t(\boldsymbol{x}|\boldsymbol{z})] \; \cdot \; p_t(\boldsymbol{x}) \\
&= v_t(\boldsymbol{x}) \, p_t(\boldsymbol{x}),
\end{aligned} \tag{12}$$

where the second line uses Bayes' rule $p_t(\boldsymbol{z}|\boldsymbol{x}) = p_t(\boldsymbol{x}|\boldsymbol{z}) \, p(\boldsymbol{z})/p_t(\boldsymbol{x})$ and the last line is the definition of $v_t(\boldsymbol{x})$. Substituting Eq. 12 into Eq. 11:

$$\frac{d}{dt} p_t(\boldsymbol{x}) = -\nabla_{\boldsymbol{x}} \cdot \big(v_t(\boldsymbol{x}) \, p_t(\boldsymbol{x})\big),$$

confirming that $(v_t, \, p_t)$ solves the continuity equation.

For the converse, we verify that $p_t(\boldsymbol{x})$ is the unique solution induced by $v_t$. Under (R1), the integrability condition required by Theorem 8.2.1(ii) of Ambrosio et al. (2008) holds:

$$\int_0^1 \int_{\mathbb{R}^d} |v_t(\boldsymbol{x})|^2 \, p_t(\boldsymbol{x}) \, d\boldsymbol{x} \, dt \; \leq \; \int_0^1 \int \int |v_t(\boldsymbol{x}|\boldsymbol{z})|^2 \, p_t(\boldsymbol{x}|\boldsymbol{z}) \, p(\boldsymbol{z}) \, d\boldsymbol{z} \, d\boldsymbol{x} \, dt \; < \; +\infty,$$

where the first inequality is Jensen's applied to $|\cdot|^2$ under $p_t(\boldsymbol{z}|\boldsymbol{x})$. The superposition principle then guarantees a probabilistic representation $p_t = e_t \sharp \eta$. Uniqueness of the solution to the continuity equation follows from Lemma 8.1.4 of Ambrosio et al. (2008) under the imposed regularity on $v_t$. Hence the marginal path induced by $v_t$ must coincide with $\int p_t(\boldsymbol{x}|\boldsymbol{z}) \, p(\boldsymbol{z}) \, d\boldsymbol{z}$.

This establishes the full consistency: the conditional vector fields $\{v_t(\cdot|\boldsymbol{z})\}_{\boldsymbol{z}}$ uniquely determine both the marginal probability path $p_t(\boldsymbol{x})$ and the marginal velocity field $v_t(\boldsymbol{x}) = \mathbb{E}_{p_t(\boldsymbol{z}|\boldsymbol{x})}[v_t(\boldsymbol{x}|\boldsymbol{z})]$, and these are consistent as a solution pair of the continuity equation. $\qquad\square$

**Remark** (Posterior collapse). *The collapsed posterior $q_t(\mathbf{z} \mid \mathbf{x}) = p(\mathbf{z})$ is a special case of any distributional family $\mathcal{Q}_{dist}$, so $\inf_{\theta, q \in \mathcal{Q}_{dist}} \mathcal{R}(\theta, q) \leq \inf_\theta \mathcal{R}(\theta, p(\mathbf{z}))$ :optimizing over $q$ can only improve upon the collapsed solution. Unlike the VAE objective, $\mathcal{R}(\theta, q)$ contains no regularization term (such as $D_{\mathrm{KL}}(q\|p)$ ) that penalizes the posterior for deviating from the prior, so there is no force driving $q$ toward collapse during optimization.*

## A.2 PROOFS IN § 3.3

Recall the posterior arising from the latent dynamic model is

$$\boldsymbol{z}_i^{[S]}|\boldsymbol{x}_i^{[S]} \sim p(\boldsymbol{z}^{[S]}|\boldsymbol{x}^{[S]}) = \prod_{s \in [S]} p(\boldsymbol{z}^s|\boldsymbol{z}^{[s-1]}, \boldsymbol{x}^{[S]}). \tag{13}$$

We present the extension of Theorem 3.1 to the latent dynamic model in the following theorem. The proof idea relies on verifying the joint continuity equation over the trajectory is satisfied and the corresponding structured conditional flow matching objective is well defined.

**Theorem A.2.** *With conditional flow defined by Eq. 7, and posterior defined by Eq. 13, the FM objective is derived to be Eq. 9, which has the same gradient as the flow matching objective that matches $v_t$ to the marginal vector field $u_t$.*

$$\mathcal{L}_{SCFM} = \mathbb{E}_{\substack{\boldsymbol{x}_1 \sim p_{data}, \boldsymbol{x}_t \sim p_t(\boldsymbol{x}_t|\boldsymbol{x}_1) \\ t \sim Unif[0,1]}} \left\| \sum_{s \in [S]} \mathbb{E}_{p_t(\boldsymbol{z}_t^{[S]}|\boldsymbol{x}_t^{[S]})} \left[ v_t(\boldsymbol{x}_t^s|\boldsymbol{z}_t^s; \theta) \right] - u_t\left(\boldsymbol{x}_t^s \mid \boldsymbol{x}_1^s\right) \right\|^2,$$

*Proof.* Assume regularity conditions that guarantee the exchange of integration and divergence, differentiation w.r.t. $t$.

We first show that $\int \sum_{s \in [S]} u_t(\boldsymbol{x}_s|\boldsymbol{x}_s^1) p_t(\boldsymbol{x}_{[S]}^1|\boldsymbol{x}_{[S]}) d\boldsymbol{x}_s^1$ is the marginalized vector field that generates $\{p_t(\boldsymbol{x}_{[S]})\}$. In the following $p(\boldsymbol{x}_1^1|\boldsymbol{x}_0) = p(\boldsymbol{x}_1^1)$ for simplicity of indexing,

$$\begin{aligned}
\frac{d}{dt}p_t(\boldsymbol{x}_{[S]}) &= \int \frac{d}{dt}p_t(\boldsymbol{x}_{[S]}|\boldsymbol{x}_{[S]}^1)p(\boldsymbol{x}_{[S]}^1)d\boldsymbol{x}_{[S]} \\
&= \int \sum_{s \in [S]} \frac{d}{dt}p_t(\boldsymbol{x}_s|\boldsymbol{x}_s^1) \cdot \prod_{j \neq s} p_t(\boldsymbol{x}_j|\boldsymbol{x}_j^1)p(\boldsymbol{x}_{[S]}^1)d\boldsymbol{x}_{[S]} \\
&= \int \sum_{s \in [S]} -\nabla \cdot (p_t(\boldsymbol{x}_s|\boldsymbol{x}_s^1)u_t(\boldsymbol{x}_s|\boldsymbol{x}_s^1)) \prod_{j \neq s} p_t(\boldsymbol{x}_j|\boldsymbol{x}_j^1)p(\boldsymbol{x}_{[S]}^1)d\boldsymbol{x}_{[S]}^1 \\
&= -\nabla \cdot \left( \sum_{s \in [S]} \int u_t(\boldsymbol{x}_s|\boldsymbol{x}_s^1)p_t(\boldsymbol{x}_{[S]}^1|\boldsymbol{x}_{[S]})d\boldsymbol{x}_{[S]}^1 p_t(\boldsymbol{x}_{[S]}) \right) \\
&= -\nabla \cdot \left( \mathbb{E}_{p_t(\boldsymbol{x}_s^1|\boldsymbol{x}_{[S]})} \left[ \sum_{s \in [S]} u_t(\boldsymbol{x}_s|\boldsymbol{x}_s^1) \right] p_t(\boldsymbol{x}_{[S]}) \right).
\end{aligned} \tag{14}$$

The second equality is by conditional independence of the transported samples for each $s \in [S]$ and applying chain rule on the product $p_t(x_s|x_s^1) \prod_{s \in [S]} p_t(\boldsymbol{x}_s|\boldsymbol{x}_s^1)$. This shows the marginal vector field is additive in the time index $s$ following the marginal vector field defined for each $s \in [S]$.

Now, we'd like to derive the conditional flow matching objective from the marginal flow matching, and show the two has the same gradient with respect to the NN parameterized marginal vector field $v_t$. The marginal VF for LDS takes the form

$$\mathbb{E}_{p_t(\boldsymbol{x}_{[S]})}\|v_t(\boldsymbol{x}_{[S]}) - u_t(\boldsymbol{x}_{[S]})\|^2. \tag{15}$$

It is then sufficient to look at the cross term and the squared term on $v_t$. Firstly,

$$\mathbb{E}_{p_t(\boldsymbol{x}_{[S]})}\langle v_t(\boldsymbol{x}), u_t(\boldsymbol{x})\rangle$$

$$= \int \left\langle v_t(\boldsymbol{x}), \sum_{s\in[S]} \int u_t(\boldsymbol{x}_s|\boldsymbol{x}_s^1)p_t(\boldsymbol{x}_s^1|\boldsymbol{x}_{[S]})d\boldsymbol{x}_s^1 \right\rangle p_t(\boldsymbol{x}_{[S]})d\boldsymbol{x}_{[S]}$$

$$= \int \sum_{s\in[S]} \int \langle v_t(\boldsymbol{x}), u_t(\boldsymbol{x}_s|\boldsymbol{x}_s^1)\rangle \int p_t(\boldsymbol{x}_{[S]}^1|\boldsymbol{x}_{[S]})d\boldsymbol{x}_{-s}^1 d\boldsymbol{x}_s^1 p_t(\boldsymbol{x}_{[S]})d\boldsymbol{x}_{[S]}$$

$$= \int \sum_{s\in[S]} \langle v_t(\boldsymbol{x}), u_t(\boldsymbol{x}_s|\boldsymbol{x}_s^1)\rangle p_t(\boldsymbol{x}_{[S]}|\boldsymbol{x}_{[S]}^1)p_t(\boldsymbol{x}_{[S]}^1)d\boldsymbol{x}_{[S]}^1 d\boldsymbol{x}_{[S]}$$

$$= \mathbb{E}_{p_t(\boldsymbol{x}_{[S]}|\boldsymbol{x}_{[S]}^1)p(\boldsymbol{x}_{[S]}^1)} \left\langle v_t(\boldsymbol{x}), \sum_{s\in[S]} u_t(\boldsymbol{x}_s|\boldsymbol{x}_s^1) \right\rangle$$

for the quadratic term, it directly follows from iterated expectations

$$\mathbb{E}_{p_t(\boldsymbol{x}_{[S]})}\|v_t(\boldsymbol{x})\|^2 = \mathbb{E}_{p_t(\boldsymbol{x}_{[S]}|\boldsymbol{x}_{[S]}^1)p(\boldsymbol{x}_{[S]}^1)}\|v_t(\boldsymbol{x})\|^2.$$

Therefore optimizing $v_t$ with Eq. 15 is equivalent to optimizing the marginal flow matching objective

$$\inf_{v_t} \mathbb{E}_{p_t(\boldsymbol{x}_{[S]}|\boldsymbol{x}_{[S]}^1)p(\boldsymbol{x}_{[S]}^1)} \left\| v_t(\boldsymbol{x}) - \sum_{s\in[S]} u_t(\boldsymbol{x}_s|\boldsymbol{x}_s^1) \right\|^2.$$

Finally, to introduce the structured FM objective with latent dynamical system, we verify the marginal vector field $v_t$ arisen from marginalizing $v_t(\boldsymbol{x}_s|\boldsymbol{z}_s)$ generates the probability path $\{p_t(\boldsymbol{x}_{[S]})\}$. The proof is similar to the previous ones, where we verify that the continuity equation is satisfied.

$$\frac{d}{dt}p_t(\boldsymbol{x}_{[S]}) = \int \frac{d}{dt}p_t(\boldsymbol{x}_{[S]}|\boldsymbol{z}_{[S]})p(\boldsymbol{z}_{[S]})d\boldsymbol{z}_{[S]}$$

$$= \int \sum_{s\in[S]} \frac{d}{dt}p_t(\boldsymbol{x}_s|\boldsymbol{z}_s) \prod_{j\neq s} p_t(\boldsymbol{x}_j|\boldsymbol{z}_j)p(\boldsymbol{z}_{[S]})d\boldsymbol{z}_{[S]}$$

$$= \int \sum_{s\in[S]} -\nabla \cdot (v_t(\boldsymbol{x}_s|\boldsymbol{z}_s)p_t(\boldsymbol{x}_s|\boldsymbol{z}_s)) \prod_{j\neq s} p_t(\boldsymbol{x}_j|\boldsymbol{z}_j)p(\boldsymbol{z}_{[S]})d\boldsymbol{z}_{[S]}$$

$$= -\nabla \cdot \sum_{s\in[S]} \int v_t(\boldsymbol{x}_s|\boldsymbol{z}_s)p_t(\boldsymbol{z}_{[S]}|\boldsymbol{x}_{[S]})d\boldsymbol{z}_{[S]} \cdot p(\boldsymbol{x}_{[S]})$$

$$= -\nabla \cdot \left( \mathbb{E}_{p_t(\boldsymbol{z}_{[S]}|\boldsymbol{x}_{[S]})} \left[ \sum_{s\in[S]} v_t(\boldsymbol{x}_s|\boldsymbol{z}_s) \right] p(\boldsymbol{x}_{[S]}) \right).$$

It is notable that the conditional independence and Markov assumption gives rise to the filtering probability $p(\boldsymbol{z}_s|\boldsymbol{x}_{[S]})$ and $p(\boldsymbol{x}_s^1|\boldsymbol{x}_{[S]})$.

In particular, the objective Eq. 9 depends on the entire sequence through the sum over $[S]$, due to the conditional independence structure of the likelihood. As it requires access to the full observed sequence at every step $s \in [S]$, the training procedure is entirely offline.

$\square$

## B EXPERIMENT DETAILS

All experiments are conducted on a MacBook Pro equipped with an Apple M2 Pro chip and 16 GB of memory. Code is available at `https://github.com/edenx/StructuredFlowAutoencoder`.

### B.1 PINWHEEL

The Pinwheel dataset is a classic benchmark for density estimation task. SFA is demonstrated to be able to meaningfully capture latent distribution as well as observed distribution.

For VAE, we use Gaussian distribution with diagonal covariance for posterior and likelihood family. The parameters of Gaussian are parameterized by MLPs and mapped from context vector. For SFA, we use MLP to parameterize the conditional vector fields $v_t$, and use Gaussian parametric family indexed by $t, x$ for the posterior flow.

With latent mixture, we use constant time Gumbel-Softmax network to model the latent class probability for both Mixture-SVAE and Mixture-SFA. We use tanh activation function for all models applied to this dataset.

To compare, the samples and latent representations are generated based on the estimated conditional probabilities,

$$\boldsymbol{z}_i \overset{i.i.d.}{\sim} p(\boldsymbol{z}), \ \boldsymbol{x}_i|\boldsymbol{z}_i \sim p_1(\boldsymbol{x}|\boldsymbol{z}_i), \ \tilde{\boldsymbol{z}}_i|\boldsymbol{x}_i \overset{i.i.d.}{\sim} q_1(\boldsymbol{z}|\boldsymbol{x}_i).$$

### B.2 SINGLE CELL RNA-SEQ

The Single Cell RNA-seq dataset Kang et al. (2018) consists of transformed count vector of size 5000, which presents challenges in modelling with CNF and likelihood based optimization. SFA directly tackles this complexity by learning a meaningful latent representation while not requiring computation of log-likelihood during training.

We parameterize the CNF $v_t$ with MLP, and uses a 32-dim Gaussian approximation family for the posterior that varies across time $t$ and observation $x$. The VAE model uses Gaussian encoder and decoder with NN parameterized parameters that is time independent. For the latent FM, we use MLP encoder and decoder, 32 dim latent space and CNF to learn the latent distribution $p(\boldsymbol{z})$.

### B.3 MNIST DATA

For Mixture-SVAE, we use Gaussian distribution with diagonal covariance for posterior and likelihood family. MLPs are used to map context vectors to the means and covariances of the Gaussians. The latent class probability is via a Gumbel-Softmax network, which uses MLP to map from context vector to logits, then apply Gumbel-Softmax trick for sampling.

For Mixture-SFA, we also use MLP to parameterize the conditional vector fields $v_t$ and Gaussian posterior with parameter indexed by $t, x$. It is notable that with larger differences in the dimensionality and scale, a linear map is used to firstly map the context vectors to vectors of the same size. Then

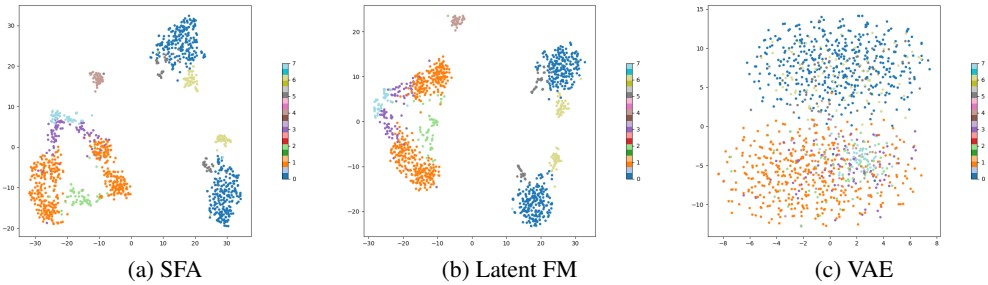

(a) SFA  (b) Latent FM  (c) VAE

Figure 5: RNAseq dataset: Latent space visualization in 2D, projected with TSNE (perplexity=30).

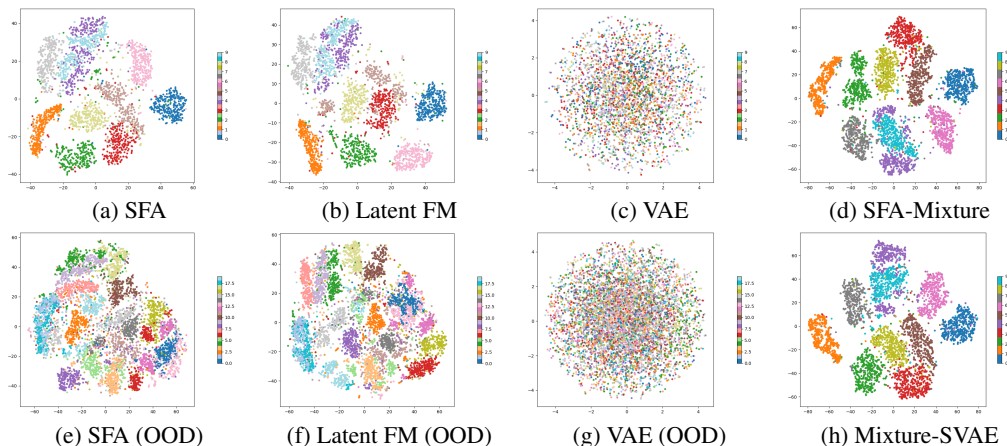

Figure 6: MNIST dataset: Latent space visualization in 2D projected with TSNE (perplexity=50). (a)-(c),(e)-(g) follows from the continuous latent graphical model, (d) and (b) employs latent mixture model. The OOD dataset consists of first 10 classes of letters and the first 10 classes of digits in EMNIST.

apply concatenation and feed to the main network. We also use Gumbel-Softmax network to model the latent class probability. Alternatively, we can use a 10 dimensional vector field to model the distribution of logits.

For both model, we use softplus activation function and train until convergence. We observe that a smaller network usually suffices for modelling the latent, which also increases training speed.

We compare the performance of the two methods from 2 perspectives.

1. **Posterior Predictive**: for every test sample $x_i$, we first sample from the posterior $z_i|x_i \sim q(z|x)$, then sample from the likelihood $x_{i,new}|z_i \sim p(x|z_i)$.

2. **Latent Space Representation**: for every test sample $x_i$, we sample from the posterior $\xi_i|x_i \sim q(\xi|x_i)$, then from $z_i|x_i \sim q(z|x)$ to obtain a latent representation of the observed data point. We visualize the class probability samples $\{\xi_i\}$ with histogram, and the sample $\{z_i\}$ with TSNE (perplexity = 30) projected from 64 dimensional space onto 3 dimensional space.

**SoftNMI** To assess the quality of latent probabilistic cluster assignment for Mixture-SFA and Mixture-SVAE, we use a soft Normalized Mutual Information (softNMI), which computes the discrepancy between a one-hot label vector and a probability vector based on entropy,

$$\text{softNMI(p, q)} = \frac{H(p) + H(q) - H(p,q)}{H(p) + H(q)} \in [0,1], \tag{16}$$

where $H(p)$ is the entropy function on the marginal, $H(p,q)$ is the entropy on the joint. Higher score suggests higher correlation between the posterior class assignment probability and true class label.

From Table 2, we observe the stochastic latent helps to increase the diversity of the generation, while variants of VAE has poor generation quality, the FM based models have better performance in image generation quality. However, there's a notable trade-off in reconstruction quality and generation diversity, where stochastic latent have higher Vendi score while suffer a slight decrease in SSIM score, and vice versa for the deterministic latent. In addition, for downstream clustering task, SFA with deterministic latent has better NMI and ARI score for within distribution sample, yet SFA with stochastic latent has better NMI and ARI for out-of-distribution samples.

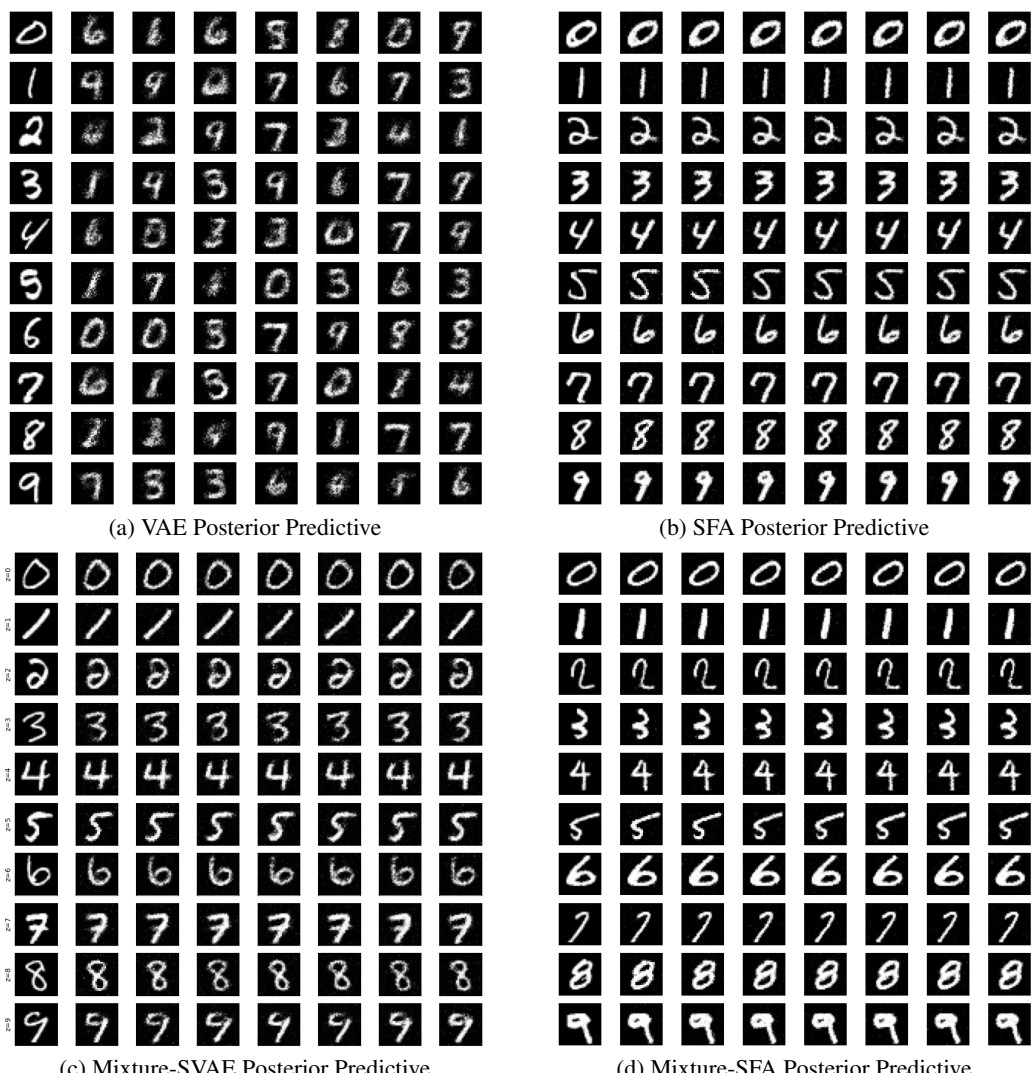

(a) VAE Posterior Predictive

(b) SFA Posterior Predictive

(c) Mixture-SVAE Posterior Predictive

(d) Mixture-SFA Posterior Predictive

Figure 7: Comparison of Posterior Predictive Results: (a) VAE, (b) SFA, (c) Mixture-SVAE and (d) Mixture-SFA.

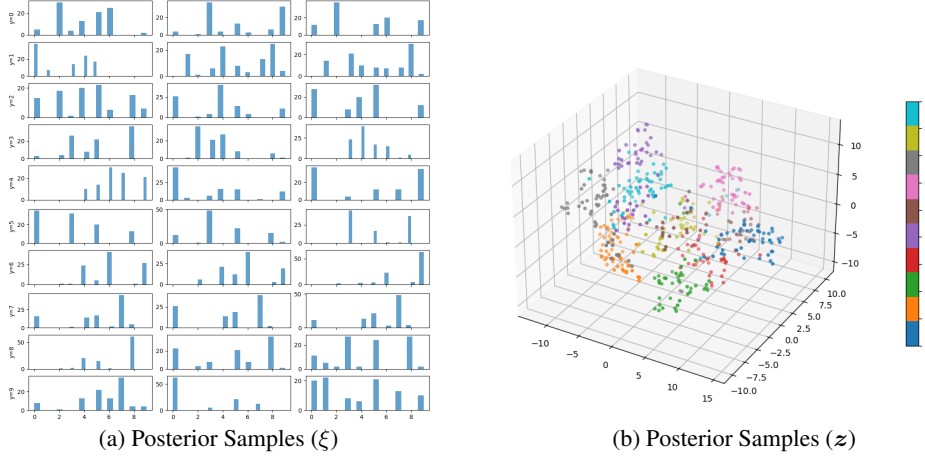

Figure 8: Mixture-SVAE Posterior Samples: (a) samples from latent variable $\xi$: each column corresponds to different images, each row corresponds to different class label; and (b) latent variable $z$, with TSNE projection from 64 dimensional to 3 dimensional space.

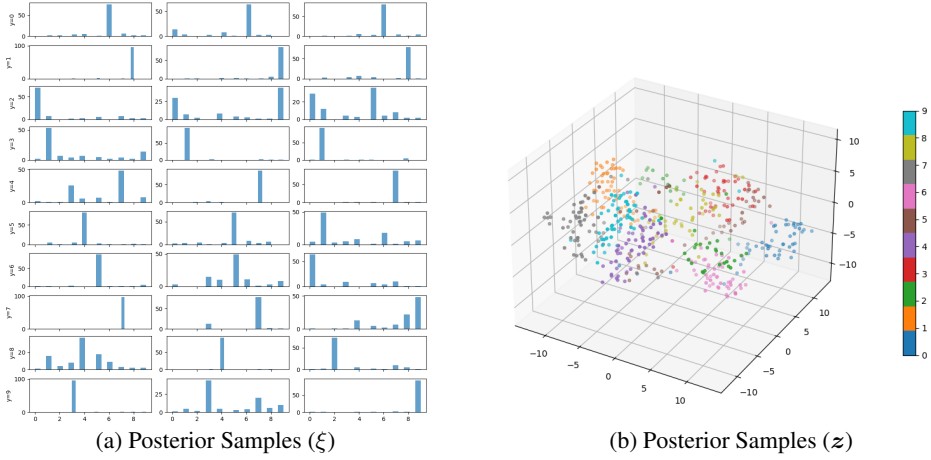

Figure 9: Mixture-SFA Results: (a) Posterior Samples ($\xi$) and (b) Posterior Samples ($z$).

## B.4 LATENT DYNAMICAL SYSTEM

To model the full conditional posterior $q(z_s|z^{[s-1]}, x^{[S]})$, we use sequence model to separately encode the observation sequence $x^{[S]}$ and full latent history $z^{[s-1]}$ to a context vector. The encoder processes each frame of $x^{[S]}$ through a shared MLP to produce per-frame embeddings, which are computed once and cached. At each autoregressive step $s$, a GRU accumulates the previously generated latents $z^{[s-1]}$ into a summary vector, which is used as a query in a cross-attention layer over all $S$ frame embeddings. This allows the model to dynamically weight which frames are most informative for producing $q(z_s|z^{[s-1]}, x^{[S]})$ adapting its attention pattern based on what latents have already been generated. The attended $x$-context is concatenated with the $z$-context and a positional scale embedding, then fed through an MLP to output the mean and variance of a diagonal Gaussian for at $s$.

Therefore, the full posterior is modeled by

$$z^s|z^{[s-1]}, x^{[S]} \sim Gaussian\left(\mu\left(h_z(z^{[s-1]}), h_x(x^{[S]})\right), \Sigma\left(h_z(z^{[s-1]}), h_x(x^{[S]})\right)\right).$$

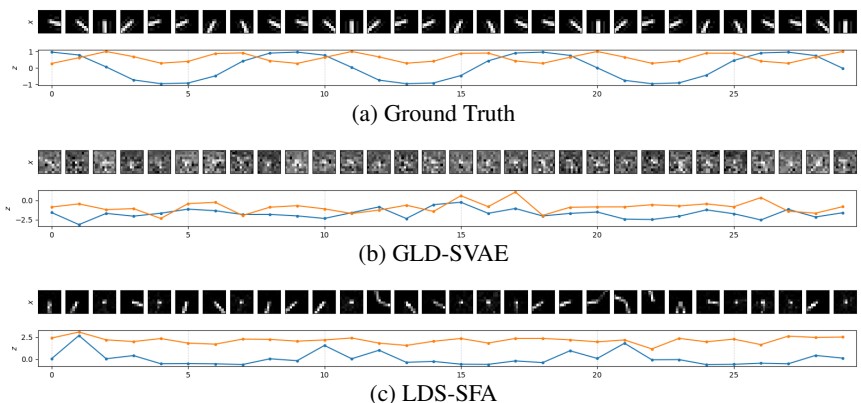

(a) Ground Truth

(b) GLD-SVAE

(c) LDS-SFA

Figure 10: Comparison of Pendulum Dynamics: Ground Truth, LDS-SFA and GLD-SVAE.

The likelihood, with conditional independence assumption is modeled by $Gaussian\left(\mu(\boldsymbol{z}_s), \Sigma(\boldsymbol{z}_s)\right)$. The likelihood is modeled by CNF, where the vector field is parameterized with MLP with SiLU activation, the incorporation of time and latent representation $\boldsymbol{z}$ is via FiLM.

We implement the probabilistic encoder VAE variant similar to the SFA counterpart, and outputs mean-field approximation. The probabilistic decoder (likelihood) also shares similar structure, however assumed to be independent Gaussian for each pixel.

## C   THE USE OF LARGE LANGUAGE MODELS (LLMS)

LLM is used to refine the code of model architecture and is used to polish the writing of the draft. LLM is not used to generate research ideas or writing to the extent that it could be regarded as a contributor.

