# OpenReview forum: "Structured Flow Autoencoders: Learning Structured Probabilistic Representations with Flow Matching"
_ICLR.cc/2026/Conference — ICLR 2026 Oral_

### Official Review · Reviewer_fJoA · 2025-10-27

**Soundness:** 3
**Presentation:** 3
**Contribution:** 3
**Rating:** 6
**Confidence:** 3

**Summary:**

This paper introduces Structured Flow Autoencoders (SFA), a new family of generative models that combine the high-fidelity density estimation capabilities of flow matching with the structured latent representation learning of probabilistic graphical models. The key technical contribution is the Structured Conditional Flow Matching (SCFM) objective, which explicitly incorporates latent variables into the flow matching framework, allowing joint learning of both the conditional likelihood p(x∣z) and the approximate posterior q(z∣x). The paper demonstrates the versatility of SFA across several domains, showing improvements in both sample quality and latent structure interpretability.

**Strengths:**

1. The paper provides a clear theoretical formulation that unifies two previously distinct paradigms: probabilistic graphical models and flow matching.

2. The Structured Conditional Flow Matching (SCFM) loss is a well-motivated extension of the flow matching objective, effectively bridging high-fidelity neural density estimation with structured latent representation learning.

3. Experiments cover toy 2D densities, images, and single-cell RNA data, demonstrating the method’s flexibility across diverse modalities and latent structural forms.

**Weaknesses:**

1. The paper discusses several posterior approximation families (time-indexed Gaussians, conditional CNFs, and Gumbel-Softmax for categorical latents )but provides only limited empirical comparison. It remains unclear how sensitive the final performance and stability are to these design choices.

2. Several parts of the pipeline still require solving ODEs (e.g., conditional CNFs), which can be computationally intensive. Providing more detailed analysis of runtime, memory usage, and overall computational complexity would strengthen the evaluation.

3. Most experiments are conducted on MNIST, Pendulum, or relatively small-scale RNA-seq datasets. It is unclear how SFA would perform on larger-scale or more complex datasets (e.g., CIFAR-10 or ImageNet). Scaling SCFM to higher-dimensional inputs or more diverse domains may introduce additional computational and optimization challenges that are not explored in this work.

4. Typos: Line 451 Table 4 --> Table 2

**Questions:**

1. As shown in Table 1, FM achieves performance comparable to SFA. It remains unclear what concrete advantages SFA provides over FM.

2. Is there a general procedure for determining the specific form of SFA when applied to different types of graphical models?

---

> ### Author Response · Authors · 2025-11-21
>
> Thank you very much for taking the time to review our work and provide detailed feedbacks. We truly appreciate your insights and are encouraged by your recognition of the framework’s strengths. Below we address the main comments and questions.
>
> **1. Posterior approximation families.**
>
> In the main paper we have introduced different graphical models that represents different latent structure that we'd like to explore for a dataset. Therefore, the choice of approximation family should be a design choice suitably chosen a-priori. Nonetheless, we refer to Appendix B in the draft for performance comparison of continuous latent and finite mixture latent on MNIST dataset with parametric approximation family.
> When it comes to using parametric family vs. continuous normalizing flow (CNF), we observe that the former tend to provide much faster runtime and sufficient estimation quality, where as CNF could be more flexible, however, requires sampling by solving ODE during training, which increases the runtime by a large margin.
> To see this empirically, we include comparison in Table A for MNIST with continuous latent. We include this update to Appendix B accordingly.
>
> We observe that CNF requires around 10 times more runtime per epoch comparing to Gaussian parametric family. In addition, the training trajectory with Gaussian parametric family is smoother, in contrast to the CNF. The performance of the two on image reconstruction and data density estimation is comparable. Therefore, with dataset like MNIST, it is sufficient to use a simple parametric family.
>
> **2. Solving ODE.**
>
> We'd like to clarify that during training, we only requires sampling from the latent distribution, whose efficiency depends on the family of model chosen. For a simple Gaussian model, the sampling speed is very fast. In addition, with the CNF likelihood, we only evaluate the vector field during training, which does not require solving the ODE. As shown in Table A, the runtime for SFA with Gaussian approximation family is comparable to VAE, whereas SFA with CNF approximation family is much slower.
>
> At test time, we first sample from the posterior then solve the ODE in order to generate samples or evaluate the likelihood. This is similar to the original FM, where at training no ODE solving is needed, but at test time it is required. This is in fact an advantage of our method, in comparison to VAE method, which requires evaluating the likelihood during training - infeasible for using models like CNF on high dimensional data.
>
> **3. SFA vs FM.**
>
> We'd like to clarify that Table 1 serves as a verification that the SFA is able to maintain the high fidelity for density estimation of observed data (like flow matching) but at the same time is able to capture the meaningful low dimensional latent structure, which enriches the class of FM generative models, and facilitates latent discovery from unstructured observations. This contribution is distinct from marginal generative model such as FM, which does not provide insight into the latent representation and structure of the data, as shown in Figure 1.
>
> **4. SFA form.**
>
>  The derivation of the loss function strictly follows from the specific graphical model for the latent, which is a model design choice. The derivation follows from identifying the conditional independence structure of the latent probability and verifying the continuity equation holds. We refer to Appendix A for the detailed procedure.
>
> Table A: Comparing CNF and Gaussian approximation family for SFA.
> | | log p(x\|z) ↑ | log p(z\|x) ↑ | Vendi ↑ | SSIM ↑ | NMI ↑ | ARI ↑ | Runtime (per epoch) |
> |---|---|---|---|---|---|---|---|
> | **VAE** | -453.648 | -85.448 | 63.286 | 0.419 | 0.039 | 0.017 | 12.789s |
> | **SFA** (Gaussian) | -916.901 | 793.262 | 25.589 | 0.716 | 0.490 | 0.356 | 13.220s |
> | **SFA** (CNF) | -907.998 | 356.141 | 23.166 | 0.654 | 0.485 | 0.355 | 167.460s |

---

> > ### Author Response · Authors · 2025-11-21
> >
> > **5. Computer Graphic Task.**
> >
> > We appreciate this suggestion and would like to clarify an important distinction regarding the scope of our contribution: this work focuses on methodological innovation rather than architectural engineering.
> >
> > The core contribution is the SCFM objective and its properties for learning structured latent representations with flow matching. This objective is architecture-agnostic and can be combined with any neural network backbone (U-Nets, transformers, etc.).
> >
> > Performance on natural images depends primarily on architectural choices (network depth, attention mechanisms, conditioning strategies, etc.), which is orthogonal to our methodological contribution. In addition, extensive architectural engineering that would obscure whether performance gains or losses stem from the SCFM objective versus architecture design decisions.
> > Our current experiments are deliberately designed to isolate and validate the methodological contribution:
> >
> > 1. Pinwheel data demonstrates that SCFM successfully learns intended latent structures while faithfully learning the data distribution;
> >
> > 2. Experiments on MNIST and Pendulum datasets show the framework handles diverse dependency patterns;
> >
> > 3. The RNA-seq dataset experiments demonstrate SFA's ability to model high-dimensional data ($5,000$ dimensions, which is higher dimensional to Cifar10 $d=32\times32\times3=3072$).
> >
> > Specific details to each can be found in Appendix B.
> >
> > **Regarding CIFAR-10**:
> >
> > Nevertheless, we recognize the value of demonstrating scalability to natural images. We are currently conducting CIFAR-10 experiments using U-Net architectures with conditioning on t and z. If completed during the revision period, we will include these results.
> >
> > However, we emphasize that the primary validation of our framework lies in: (1) theoretical soundness (Propositions 3.1-3.2, Theorem 3.3), and (2) empirical demonstration that SCFM achieves its design goal. Our current experimental suite already provides this validation across multiple data modalities and structural dependencies.
> >
> > ---
> > ---
> >
> > Lastly, we have addressed the typos in the updated draft.
> >
> > We believe our responses, together with the additional experimental results in Table A, comprehensively address your concerns about posterior approximation choices, and the advantages of SFA over standard Flow Matching. We are committed to incorporating all your valuable suggestions in the revised manuscript, including enhanced clarity on the computational analysis and more explicit comparison of design choices.
> >
> > We hope these clarifications strengthen your confidence in our contribution and look forward to your final assessment.

---

### Official Review · Reviewer_KF7s · 2025-10-30

**Soundness:** 3
**Presentation:** 2
**Contribution:** 3
**Rating:** 4
**Confidence:** 3

**Summary:**

The authors develop a structured conditional flow matching objective which is used to train their Structured Flow Autoencoder architecture. Likelihoods are similar to conditional continuous normalizing flow frameworks. Their aim is to achieve similar quality of samples to standard flow matching while retaining latent structure. They do this by jointly learning the conditional continuous normalizing flow likelihood and the approximate posterior.

**Strengths:**

Strengths
A pretty comprehensive suite of small-to-medium scale experiments, with a toy Pinwheel dataset, MNIST, RNA, and a video dataset.

Their method seems to have the fidelity of the flow-based models while keeping the latent structure intact as with the VAEs.

The structure seems to be well preserved when visualizing the MNIST digits with TSNE, wit SFA-Mixture, SFA, Latent-FM, and GMVAE giving similar groupings for the classes.

**Weaknesses:**

Weaknesses:

There isn't really one consistent SFA model for testing on the different problems, with SFA, Mixture-SFA, and LDS-SFA, etc. Is there a necessity for there to be modifications to be different for every data set?

Also some terms like LDS-SFA is not properly defined. I can infer that it is SFA on the LDS dataset, but hard to grasp what makes it different.

**Questions:**

The log likelihood in Table 2 and Table 3 for SFA has value 725.232 and 384.137, which looks surprisingly large. How is this calculated, and any idea why the density is so concentrated?

---

> ### Author Response · Authors · 2025-11-21
>
> Thank you very much for taking the time to review our work and provide detailed feedbacks. Below we address the main comments and questions.
>
> **1. Consistent SFA model.**
>
> The core SFA framework is consistent across all experiments. The fundamental methodology is captured in Equation 6 (the SCFM objective) and Proposition 3.1, which applies uniformly to any graphical model:
>
> $$
> \text{SCFM} = \mathbb{E}\left[\left\|\mathbb{E}\_{p_{t}(z_t|x_t)}[v_{t}(x_t|z_t;\theta)]-u_t(x_t | x_1)\right\|^2\right]
> $$
>
> where the outer expectation is over $x_1 \sim p_{\text{data}}(x_1)$, $x_t \sim p_t(x \mid x_1)$, and $t \sim \text{Unif}[0,1]$.
>
> This objective remains unchanged across all our experiments. What differs is the **choice of latent graphical structure**, which is a design decision based on the data characteristics and modeling goals—exactly analogous to how VAE variants (vanilla VAE, GMVAE, sequence VAE) share the same ELBO objective but employ different latent structures [1,2,3].
>
> **Why different latent structures are necessary:**
>
> Just as there is no single universal VAE architecture for all data types, SFA requires appropriate latent structures for different applications:
>
> - SFA (continuous latent): Standard latent variable model for general representation learning (analogous to vanilla VAE)
>
> - Mixture-SFA: Automatically discovers cluster structures in data (analogous to GMVAE)
>
> - LDS-SFA: Captures temporal dynamics in sequential data (analogous to sequence VAE/RNN-VAE)
>
> The framework is consistent and the instantiation is flexible. This flexibility is a strength, allowing practitioners to choose appropriate structural assumptions for their domain. We will clarify this distinction more explicitly in the revised manuscript to avoid confusion.
>
> [1] Johnson, M. J., Duvenaud, D. K., Wiltschko, A., Adams, R. P., & Datta, S. R. (2016). Composing graphical models with neural networks for structured representations and fast inference. Advances in neural information processing systems, 29.
>
> [2] Dilokthanakul, N., Mediano, P. A., Garnelo, M., Lee, M. C., Salimbeni, H., Arulkumaran, K., & Shanahan, M. (2016). Deep unsupervised clustering with gaussian mixture variational autoencoders. arXiv preprint arXiv:1611.02648.
>
> [3] Lin, W., Hubacher, N., & Khan, M. E. (2018). Variational message passing with structured inference networks. arXiv preprint arXiv:1803.05589.
>
>
> **2. Log likelihood evaluation.**
>
> The log-likelihood for $q(z_1|x_1)$ is computed by firstly obtain posterior sample $z_1\sim q(z_1|x_1)$, then evaluate with `torch.distributions` built-in log-likelihood function; for the observed conditional $p(x_1|z_1)$, its log-likelihood is evaluated based on solving adjoint ODE [4]. We use the same evaluation for VAE variants.
> Therefore the concentration of the likelihood for SFA suggests the low dimensional latent representation $z$ captures the corresponding data $x$ very well. This is actually evidence of SFA's success: improving both latent representation quality and density estimation compared to VAE variants.
>
> [4] Chen, R. T., Rubanova, Y., Bettencourt, J., & Duvenaud, D. K. (2018). Neural ordinary differential equations. Advances in neural information processing systems, 31.
>
> **3. Notation clarification.**
>
> - LDS-SFA = Structured Flow Autoencoder with Latent Dynamical System structure
>   - Used for sequential data (Pendulum experiments)
>   - Incorporates temporal Markov dependencies: $p(z_t|z_{t-1})$
>   - Section 3.3
>
> - Mixture-SFA = SFA with finite mixture latent structure
>   - Used for data with cluster structure (MNIST digit classes)
>   - Incorporates categorical latent: $p(\xi) = Categorical(\pi)$
>   - Section 3.2
>
> - SFA (without modifier) = SFA with continuous latent structure
>   - Standard encoder-decoder with continuous latent variables
>   - Section 3.1
>
> We will add a nomenclature table in the revised manuscript for easy reference.
>
> ---
> ---
>
> We sincerely thank you for your constructive feedback and careful review. We recognize that our presentation may have obscured the consistency and generality of the SFA framework, and we will revise the manuscript to address these clarity issues explicitly. We hope our responses demonstrate that:
>
> 1. The core SFA methodology is consistent and principled (captured by the SCFM objective
>    in Equation 6)
>
> 2. The flexibility to handle different latent structures is by design and analogous to
>    the VAE family
>
> 3. The high log-likelihood values reflect SFA's success in achieving our design goal
>
> We respectfully ask for a favorable re-evaluation in light of our clarifications. We believe the issues you identified can be effectively addressed in revision, and that the methodological contribution — a principled framework for learning structured latent representations with flow matching—represents a valuable addition to the field.
>
> We are committed to incorporating all your suggestions in the revised manuscript and
> would greatly appreciate your reconsidered assessment.

---

> > ### Comment · Reviewer_KF7s · 2025-11-25
> >
> > Thanks for the clarifications. I have increased my score to a 6 accordingly.

---

### Official Review · Reviewer_EKhV · 2025-10-31

**Soundness:** 3
**Presentation:** 3
**Contribution:** 3
**Rating:** 8
**Confidence:** 2

**Summary:**

This paper introduces structured flow autoencoders, a method that combines the advantages of flow matching (generating high-resolution diverse data ) and VAEs (learning structured latents for the data).  The idea is to jointly learn the likelihood model p_t(.|z) and the posterior p_t(.|x) using a flow matching objective, where the marginal velocity vector v_t(x) is replaced by an expectation over conditional velocity vectors v_t(x | z) under p_t(z | x).  The parametrization of the posterior  p_t(z | x) can be done either with a CNF or with simple parametric families e.g. gaussians.  They introduce several possible generative processes for x | z, either  continuous latent model, latent mixture model, or latent dynamic system, and such a process is set before training.
They run experiments on pinwheel, mnist, rna-seq and pendulum video data, and show that their method shows improvements compared to the baselines both in terms of (i) the structure of the learned latent space and (ii) the likelihood p(x|z) and the posterior p(z | x).

**Strengths:**

- very nice and original idea to learn the latent structure with a flow matching objective instead of a VAE and theoretically grounded with the conditional vector field equation,
- Clarity and mathematical / scientifc rigor of of the paper,
- Solid and diverse set of experiments and of evidence,  and diverse set of baselines , nice 2D visualisations.

**Weaknesses:**

- Regarding the novelty of the eq.5, used in proposition 3.1 : this conditional vector field equation was introduced before in https://arxiv.org/pdf/2302.00482v1 eq 9 (however they are not conditioning on latents, but on the unnoised sample x_1)

-  Regarding the clarity of the toy experiments: Although the theory for the SFM equation is clear, I wasn't able to understand the nature of the learned latent variables for the pinwheel dataset and MNIST dataset

**Questions:**

Hello authors,
First, I'd like to start by acknowledging that I couldn't spend as much time as I wanted to on your paper to fully understand the theory / baselines, and that there are some parts for which the theory is a bit vague for me, so apologies in advance if some of the questions/claims are evident.

- Figure 1 : I don’t fully understand this figure. You train without the color and you want the learned latent variable to be the color?
- For MNIST / Pinwheel: What is the structure of sampled the latent variable in this example? is it  a predicted  (mixture of ) mean and variance, and I guess they are one-dimensional here? If not what are their dimensions?
- What do the intermediate-time latents z_t  look like, e.g. for MNIST or for pinwheel data ? are they interpretable?
- In the paragraph "SVAE and SFA comparison", you claim that SFA has an advantage by jointly learning p(x|z) and p(z|x) , whereas SVAE has a "generation-latent learning trade-off". I don't understand this, could you elaborate more on it and why it is a drawback compared to SFA?
- Could you explain the advantage of your method compared to latent flow matching with structured autoencoders? What is the advantage of also modeling z_t with time, instead of just learning the encoder/decoder for clean data as in latentFM?

---

> ### Author Response · Authors · 2025-11-21
>
> Thank you very much for taking the time to review our work and provide thoughtful, detailed feedback. We truly appreciate your insights and are encouraged by your recognition of the framework’s novelty and strengths. Below we address the main comments and questions.
>
> **1. Related work.**
>
> We acknowledge the referenced paper [1] as related work. However, the focus of the paper [1] is to develop alternative transport maps to the original FM formulation. It's contribution lies within proposing various probability path formulated with different optimal transport problems, such as Schr\"odinger bridge (conditioning on both the start and the end point). The goal and use-case of this paper is very different from ours. Our method composes latent structure to the marginal flow matching (FM), which enriches the class of FM model from a representation learning/latent variable model perspective, as well as from the perspective of learning conditional paths in probability space.
>
> [1] Tong, A., Malkin, N., Huguet, G., Zhang, Y., Rector-Brooks, J., Fatras, K., ... & Bengio, Y. (2023). Conditional flow matching: Simulation-free dynamic optimal transport. arXiv preprint arXiv:2302.00482, 2(3).
>
> **2. Latent variable.**
>
> In Figure 1, the training data are samples from the Pinwheel dataset without labels. SFA then learn both the 1D latent $z$ that should capture the rotational angle of the fins, as well as accurately recovering the data density, which is what Figure 1 shows. In particular, the color ($z$) are given by random samples from the base distributions, which is conditioned to generate the 2D sample points $x$. The distribution of $z$ corresponds to the learned posterior approximation distribution $q(z|x)$.
>
> We include additional visualization of latent trajectory in Appendix B. In particular, with MNIST dataset we visualize trajectory of $z_t$ by sampling from $q(z_t|x)$ at each time point $t$. In this case, the latent should roughly  corresponds to the class of the digit. We observe that the latent $z_t$ of different digits 0-9 moves to become more separated (for the 1st dimension of PCA projected trajectory), which suggests that our latent adapts from base distribution $N(0,1)$ to conditional distribution informed by observed $x$. In addition, $z_t$ helps to anchor the trajectory of $x_t\sim p(x_t|z)$ as well, and we requires lower dimensional embedding of $t$ comparing to marginal FM.
>
> **3. SVAE vs SFA.**
>
> SVAE suffer from posterior collapse (posterior is not learning any structure in the data and reduces to prior) when using more complex model, and requires careful tuning of the regularization parameter in balancing the learning of likelihood and posterior (as in $\beta$-VAE). However, for SFA, we incorporate the trajectory of the latent which help to anchor the trajectory of the likelihood, where the time indices does not capture the majority of the information but the latent does. This intuition is why our method is able to capture the latent structure while provide high fidelity to data generation with complex model class for the likelihood.
>
> **4. SFA vs Latent FM.**
>
> From the experiments included in the draft, SFA is most suitable when the latent dimension is assumed to be low, and when one is interested in representation learning as well as density estimation. Most notably, SFA is very good at learning the data density while capturing meaningful latent representation. One example is the Pinwheel dataset, where the latent FMs (with deterministic and probabilistic encoder decoder) are not able to reconstruct the data density well, whereas SFA excels at capturing both the observed data distribution and the latent distribution. Therefore, joint learning of likelihood and posterior in matching the marginal is the preferred paradigm for density estimation tasks. Latent FM on the other hand would be suitable for tasks such as reducing training and generation time without concerns for accurate latent structure and density estimation, as it directly learn the trajectory in a much lower dimensional space. In this case, the geometry of the latent space is suggested by the choice of encoder and decoder, but not informed by the marginal distribution of the observed.
>
> ---
> ---
>
> Thank you again for your time and feedback. We hope the rebuttal clarifies your questions and addresses your concerns. We look forward to your final assessment of our work.

---

### Official Review · Reviewer_AJKg · 2025-11-03

**Soundness:** 3
**Presentation:** 3
**Contribution:** 2
**Rating:** 6
**Confidence:** 3

**Summary:**

The authors propose Structured Flow Autoencoders (SFA), a framework that augments Continuous Normalizing Flows (CNFs) with graphical model structures.  The paper demonstrates the versatility of SFA across multiple settings, including models with continuous and mixture latent variables as well as latent dynamical systems. Experimental results show that SFA achieves superior data fitting performance while preserving meaningful and structured latent representations.

**Strengths:**

The paper is clearly written and well organized, making it easy to follow. The proposed model and its accompanying theoretical analysis are sound and logically developed.

**Weaknesses:**

Despite its clarity, the paper presents several limitations that the authors should address:

(a) The proposed approach essentially represents a straightforward combination of Flow Matching and Variational Autoencoders (VAEs). As such, the conceptual novelty appears limited, and the contribution may be incremental.

(b) The probabilistic structures considered are restricted to latent variable models and latent chain models. Given this narrow scope, the term structured probabilistic models may be overstated; the work would be more accurately described as focusing on latent models with specific dependencies.

(c) The experimental evaluation lacks results on natural image datasets, which are standard benchmarks for assessing the scalability and expressiveness of flow-based models. Including such experiments—and comparing SFA directly with Flow Matching and VAE baselines—would significantly strengthen the empirical claims.

**Questions:**

1. How does SFA perform in high-dimensional domains, such as image or video modeling?

2.Can the proposed structured latent representations be extended to discrete or hybrid graphical models?

---

> ### Author Response · Authors · 2025-11-21
>
> Thank you very much for taking the time to review our work and provide detailed feedback. Below we address the main comments and questions.
>
> **1. Novelty.**
>
> SFA is a novel methodology in composing flow matching with latent graphical model, which uses continuous normalizing flow (CNF) for density estimation and generation. SFA represents a fundamentally different methodology from combining VAE with flow matching, despite superficial similarities in learning encoders and decoders.
>
> The formulation of VAE relies on minimizing the negative ELBO function
> $$ELBO = \mathbb{E}\_{q(z|x)}[\log p(x|z)] - KL(q(z|x)\|p(z))$$
> whereas our method depends on the proposed latent aware flow matching objective SCFM
> $$
> \text{SCFM} = \mathbb{E}\left[\left\|\mathbb{E}\_{q_{t}(z_t|x_t)}[v_{t}(x_t|z_t;\theta)]-u_t(x_t | x_1)\right\|^2\right]
> $$
>
> where the outer expectation is over $x_1 \sim p_{\text{data}}(x_1)$, $x_t \sim p_t(x \mid x_1)$, and $t \sim \text{Unif}[0,1]$.
> The idea of approximating posterior distribution is a Variational Bayes paradigm that is one of the foundational idea in modern Bayesian inference, which we leverage in our work as well. However, this does not equate our work to VAE with flow matching.
>
> Instead, a straightforward combination of VAE and flow matching would be using ELBO as the loss function and continuous normalizing flow as the likelihood model. This approach would suffer from posterior collapse (reduction to prior) owing to the class of CNF model being too flexible and the latent would be ignored during training. On the other hand, our proposed SFA excels at learning both the data density and latent representation, as demonstrated with the examples in the current draft (more in Appendix B).
>
> **2. Scope.**
>
> We'd also like to point out that the proposed methodology provides a general formulation of incorporating latent structures to the otherwise marginal flow matching model, and we presented the three graphical models (continuous, finite mixture and dynamics latent structure) as three representative examples of common graphical structures. The propositions and theorems developed can be adapted to any graphical models of choice. In fact Proposition 3.1 is formulated for arbitrary graphical models with dependencies between latent variables z and observations x, regardless discrete or continuous nature of $z$. The structure within the latent could be arbitrary as it would be integrated out before linking to $x$. Therefore the framework naturally extends to: (a) hierarchical latent structures, (b) discrete latents via Gumbel distributions.
>
> We chose these three examples because they are (1) widely applicable across different domains, (2) representative of different dependency types (continuous, finite mixture, temporal), and (3) sufficient to demonstrate the framework's flexibility. We will clarify this in the revised manuscript.
>
> **3. Natural image benchmarks.**
>
> We appreciate this suggestion and would like to clarify an important distinction regarding the scope of our contribution: this work focuses on methodological innovation rather than architectural engineering.
>
> The core contribution is the SCFM objective and its properties for learning structured latent representations with flow matching. This objective is architecture-agnostic and can be combined with any neural network backbone (U-Nets, transformers, etc.).
>
> Performance on natural images depends primarily on architectural choices (network depth, attention mechanisms, conditioning strategies, etc.), which is orthogonal to our methodological contribution and would obscure whether performance gains or losses stem from the SCFM objective versus architecture design decisions.
>
> **Our experimental validation strategy:**
>
> Our current experiments are deliberately designed to isolate and validate the methodological contribution:
>
> 1. Pinwheel data demonstrates that SCFM successfully learns both latent structures and the data distribution;
> 2. Experiments on MNIST and Pendulum datasets show the framework handles diverse dependency patterns;
> 3. The RNA-seq dataset experiments demonstrate SFA's ability to model high-dimensional data (5,000 dimensions).
>
> Details can be found in Appendix B.
>
> **Regarding CIFAR-10:**
>
> Nevertheless, we recognize the value of demonstrating scalability to natural images. We are currently conducting CIFAR-10 experiments using U-Net architectures with conditioning on t and z. If completed during the revision period, we will include these results.
>
> However, we emphasize that the primary validation of our framework lies in: (1) theoretical soundness (Propositions 3.1-3.2, Theorem 3.3), and (2) empirical demonstration that SCFM achieves its design goal. Our current experimental suite already provides this validation across multiple data modalities and structural dependencies.
>
> ---
> ---
>
> We hope the rebuttal addresses your main questions and concerns and supports a favorable re‑evaluation. Thank you again for your time and feedback.

---

### Meta-Review · Area_Chair_hVxy · 2026-01-06

**Summary:**

This paper introduces Structured Flow Autoencoders (SFA), an approach which develops a new flow-matching based objective (SCFM) which extends FM to incorporate latent variables, enabling simultaneous learning of likelihood and posterior for probabilistic latent variable models. The new method is demonstrated for continuous latent variables, finite mixture models, and latent dynamical systems and is shown to outperform VAEs.

While there was a concern of limited conceptual novelty relative to VAEs, all reviewers appreciated the contributions of the paper and unanimously recommended acceptance.

**Reviewer Concerns:**

There were initial concerns regarding experimental evaluation and computational complexity -- these were addressed in rebuttal, clarifying that a consistent framework was used in all evaluations and that ODE solving was only required for generation and not training.

The reviewers pointed out conceptual similarity to the VAE framework, but acknowledged important differences in the objective used (SCFM vs ELBO) and inference.

**Reviewer Scores:**

All reviewers appreciated the contributions of the paper and unanimously recommended acceptance. I suspect they are unlikely to have substantially changed their scores.

---

### Decision · Program_Chairs · 2026-01-26

Accept (Oral)